# Mineralogical Characteristics of Biotite and Chlorite in Zuluhong Polymetallic Deposit: Implications for Petrogenesis and Paragenesis Mechanisms of the Tungsten and Copper

**Rui Cao [1,2], De-Fan Chen [2,*], Hao-Dong Gu [3], Bin Chen [4] and Sheng-Chao Yan [5]**

1    State Key Laboratory of Geohazard Prevention and Geoenvironment Protection, Chengdu University of Technology, Chengdu 610059, China
2    College of Earth Science, Chengdu University of Technology, Chengdu 610059, China
3    State Key Laboratory of Biogeology and Environmental Geology, China University of Geosciences, Wuhan 430074, China
4    Department of Earth and Space Sciences, Southern University of Science and Technology, Shenzhen 518055, China
5    State Kay Laboratory of Lithospheric Evolution, Institute of Geology and Geophysics, Chinese Academy of Sciences, Beijing 100029, China
*    Correspondence: defanchencdut@163.com

**Abstract:** The Zuluhong quartz-vein-type polymetallic deposit, located in the Alatau area of Western Tianshan, China, is a particular and typical tungsten deposit associated with copper. This paper presents major and trace element analyses of magmatic and altered (i.e, chloritized) biotite from the deposit, in order to identify the source of the magmas and characterize the mineralization physical-chemical condition. Magmatic biotite is Fe-rich and has high Rb/Ba ratios (0.27–9.14), indicative of extensive differentiation of granite. Moreover, magmatic biotite has total rare earth element (∑REE) contents that are 5–10% of the whole-rock contents, shows slight light REE depletion, and negative Ce anomalies. Magmatic biotite is enriched in some large-ion lithophile elements (LILE; e.g., Rb and K) and depleted in some high-field-strength elements (HFSE; e.g., Th and Nb). These geochemical features, coupled with geological evidence, indicate that the Zuluhong intrusion is a highly fractionated I-type granite derived from lower crustal melting. During ore formation, magmatic biotite was partially to totally altered to chlorite due to interaction with ore-forming fluids. The temperature and oxygen fugacity decreased during alteration. The mineralization in the Zuluhong polymetallic deposit can be divided into at least two stages. In the early stage, quartz-vein-type wolframite mineralization formed from Si- and volatile-rich fluids that were derived from fractionated granitic magma. In the later stage, W–Cu ores formed as metal sulfides were dominated by chalcopyrite. The later ore-forming fluids experienced a decrease in temperature and oxygen fugacity as they reacted (i.e, chloritization and lesser silicification) with reducing wall rocks around the contact zone of the intrusion.

**Keywords:** Zuluhong; W–Cu mineralization; biotite; oxygen fugacity; highly fractionated I-type granite

## 1. Introduction

The Alatau area in the Western Tianshan, China, is located at the southwestern margin of the Central Asian Orogenic Belt (CAOB). This region has experienced numerous late Paleozoic magmatic events [1], as well as large-scale Cu polymetallic mineralization [2,3]. The Zuluhong W–Cu polymetallic deposit is a newly discovered deposit in the Alatau area. However, it is generally considered that W and Cu do not coexist in the same mineralization area [4], due to their different ore-forming processes [5,6], source magmas [7–10], and ore-forming fluids [11–13]. Few previous studies have examined the Zuluhong deposit, focusing mainly on the age and petrology of the associated granite [14].

The nature of the late mineralization fluid and changes in oxygen fugacity during magma evolution in the Zuluhong deposit have not been studied. As such, the mechanism of W–Cu paragenesis remains unclear.

Biotite is a significant ferromagnesian mineral in granite, and records both the nature and physicochemical conditions (e.g., temperature and oxygen fugacity) of the granite magma from which it formed [15–20]. Chlorite is a ubiquitous product of fluid–rock interactions in hydrothermal systems, and can be used to constrain the changing temperature and oxygen fugacity conditions during alteration [21–24]. Physicochemical conditions and fluid evolution can be revealed through studies of different types of biotite, particularly magmatic and altered biotite. For example, through the study of biotites in altered rocks, Yuguchi et al. [25] constrained the pH and redox evolution during alteration. Similarly, a biotite study of the Darreh–Zar porphyry Cu deposit by Parsapoor et al. [26] revealed the temperature and pressure changes during alteration, as well as the volatile chemistry of the fluids involved.

Magmatic biotites in the Zuluhong granite body are euhedral with a high degree of crystallinity, but around mineralized quartz veins, alteration is strong and dominated by the chloritization of biotite. This study investigates the magma source and evolution, fluid properties, and oxygen fugacity changes during alteration, through major and trace element analyses of magmatic and altered biotite in the Zuluhong deposit. The results provide new perspectives on metallogenic models of tungsten deposit associated with copper.

## 2. Regional Geological and Mineralization Background

The Alatau area was a Paleozoic active continental margin in the Yili Plate between the Junggar and Tarim plates and was an important component of the Eurasian Central Asia–Mongolia–Okhotsk arc orogenic belt (Figure 1a). The basement rocks comprise mainly Paleozoic strata, including the upper Devonian Tuosikuertawu, lower Carboniferous Aksha, upper Carboniferous Dongtujinhe, and lower Permian Wulang formations. Precambrian basement has not been located in this region. Late Paleozoic magmatism in this area led to the emplacement of intermediate–silicic hypabyssal rocks, which were closely related to the mineralization. The intrusive rocks are mainly dikes and stocks. Metamorphic rocks are not well-developed in this area, apart from hornfels in contact metamorphic zones between the intrusions and wall rocks. The area is crossed by several dextral strike-slip and thrust faults, and folds are only found in the lower Carboniferous Aksha Formation.

A series of Cu polymetallic deposits (e.g., the Lamasu Cu–Zn and Kekesai, Dongtujin, and Dabate Cu–Mo deposits) (Figure 1b) occur in the Alatau area. The Zuluhong W–Cu polymetallic deposit has been found recently, located at 81° E and 45°N, 34 km north of Wenquan County to the north of Xinjiang [27]. The strata in the Zuluhong mine are mainly the lower Carboniferous Aksha Formation, comprising micrites, sandstones, siltstones, and slates. The Zuluhong intrusion are composed of monzogranite and granodiorite in the mining area and are formed at 320 ± 2 Ma and 309 ± 1 Ma, respectively [28]. The intrusion was emplaced into early Carboniferous strata, has an irregular oval shape, and extends over an area of ~10 km². The intrusion is zoned from a marginal facies of porphyritic fine-grained biotite monzogranite to an internal facies of porphyritic medium-grained biotite monzogranite. The structure of the mining area includes a main NE–SW-striking thrust fault, which controls the distribution of the intrusion and orebody (Figure 1c).

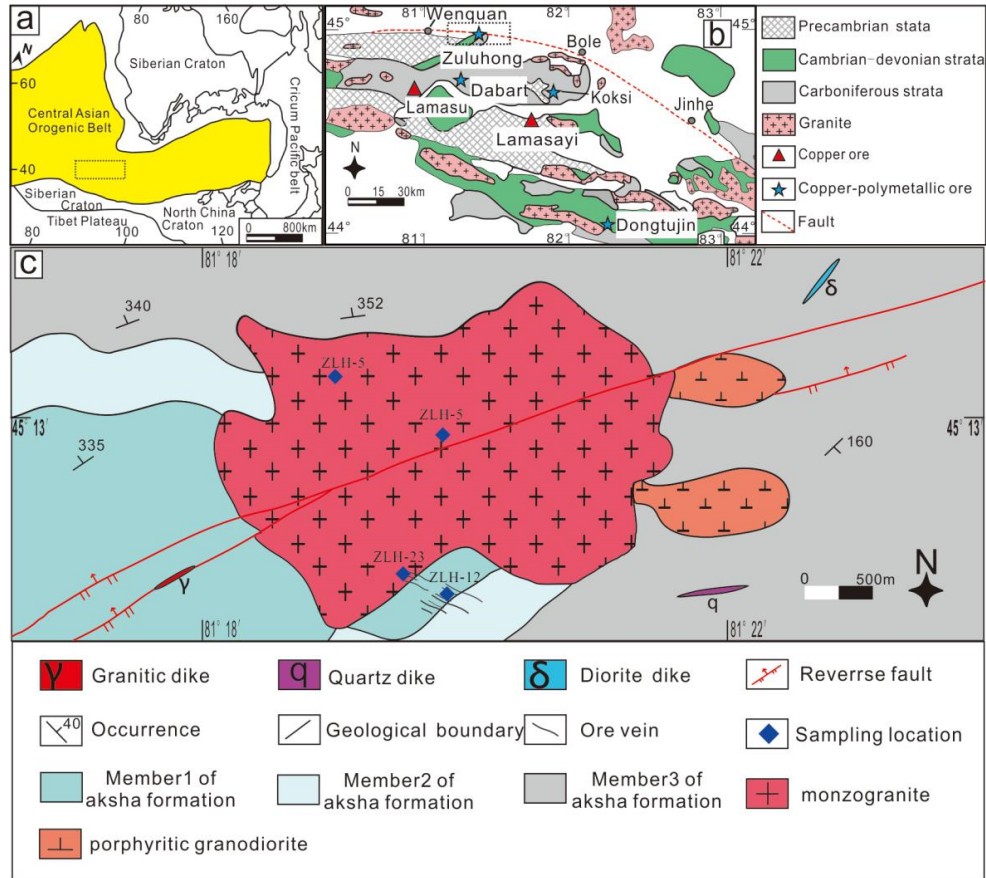

**Figure 1.** Geological sketch map of the Central Asian Orogenic Belt in the Alatau area and the Zuluhong W–Cu deposit, showing (**a**) the location of the Central Asian Orogenic Belt (modified after Jahn et al. [29]), (**b**) the location of the Zuluhong and northwest Tianshan metallogenic belt, and (**c**) the distribution of stratigraphic units and major structures in the Zuluhong W–Cu deposit.

The Zuluhong Granite can be divided into a monzogranite and porphyritic granodiorite from its core to rim. Several ore minerals are found in different types of mineralized quartz veins, which allows the veins to be divided into early and late stages. The main minerals in the early quartz veins are wolframite and quartz. Wolframite is mostly 2–5 cm and up to 8 cm in size, euhedral, and occurs as aggregates or single crystals (Figure 2a). The contacts between wolframite and quartz are smooth without a metasomatic texture (Figure 2b), indicating that wolframite and quartz co-crystallized. The mineralogical features of the late quartz veins are more complex than those of the early veins. The main minerals in the late quartz veins are wolframite, chalcopyrite, pyrite, and quartz. The characteristics of wolframite are similar in the two types of veins. Chalcopyrite (Figure 2c,d) and pyrite (Figure 2e) replace wolframite or infill voids and cleavage planes (Figure 2f). Extensive alteration is evident around the late quartz veins, mainly chloritization. As such, the two major mineralization stages were a quartz–wolframite stage and a sulfide stage. Biotite was altered to chlorite during the second stage.

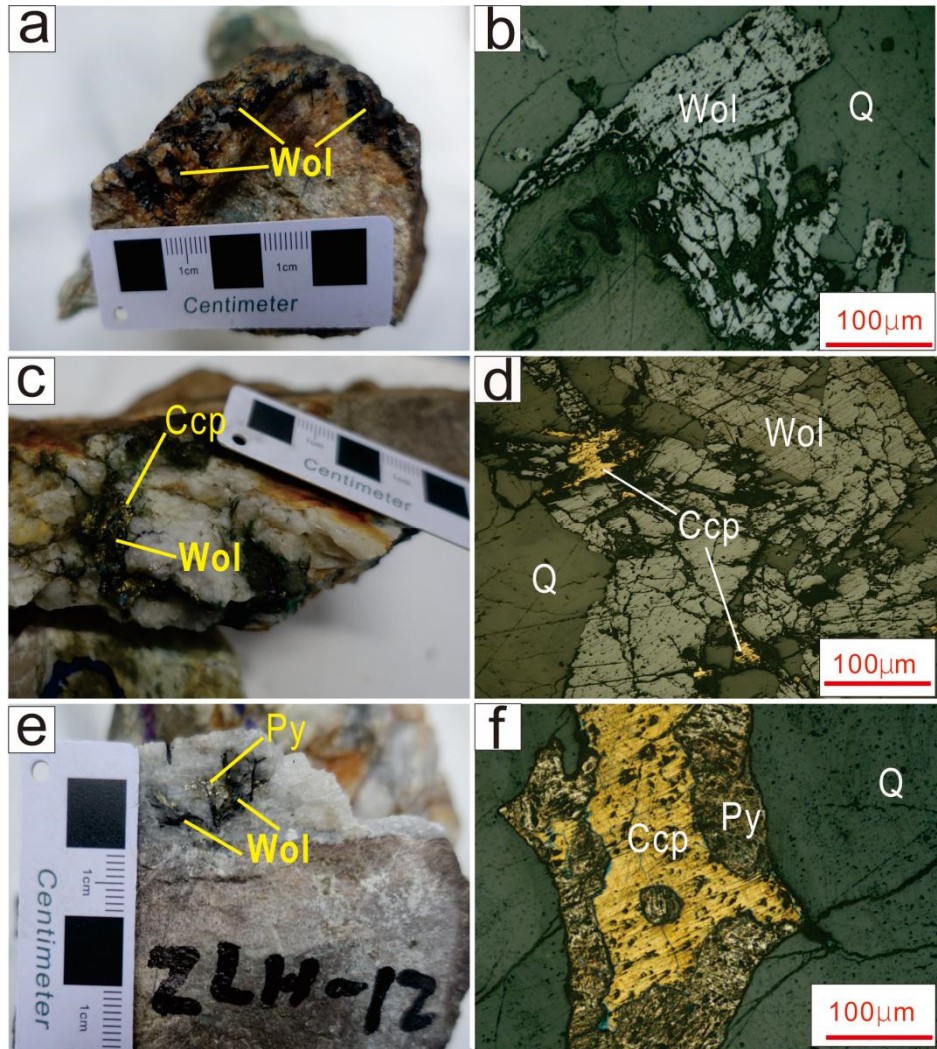

**Figure 2.** Typical examples of mineralized quartz veins in the Zuluhong deposit. (**a**) Platy wolframite (2–5 cm across) in an early quartz vein. (**b**) Smooth contact between wolframite and quartz (plane-polarized light). (**c**) Chalcopyrite replacing wolframite and infilling cleavage planes. (**d**) Chalcopyrite replacing wolframite in a late quartz vein (plane-polarized light). (**e**) Pyrite replacing wolframite in a late quartz vein. (**f**) Pyrite infilling voids and cleavage planes, surrounded by quartz (plane-polarized light). Ccp = chalcopyrite; Py = pyrite; Q = quartz; Wol = wolframite.

## 3. Analytical Methods

Ten thin sections of the Zuluhong Granite were studied using a microscope (ECLIPSE LV100N POL). EDX map of the biotite was collected at the School of Earth Sciences, Chengdu University of Technology, Chengdu, China, using a Nova 450 field-emission scanning electron microscope at a voltage of 30 kV. Major element compositions of biotite were acquired at the School of Earth Sciences and Resources, Chinese University of Geology, Wuhan, China, using a JEOL JXA-8230 electron microprobe analyzer (EMPA) operated at a voltage of 15 kV, beam current of 20 nA, spot size of 1μm, and maximum counting time of 20 s. SPI standard minerals were used as standards. X-ray intensities were corrected using the ZAF methodology. Analytical errors on the major element determinations are estimated to be ~1% relative.

Trace element compositions of biotite were determined by laser ablation–inductively coupled plasma–mass spectrometry (LA–ICP–MS) at the School of Resources and Environmental Engineering, Hefei University of Technology, Hefei, China. The analyses were conducted with a GeoLas193 LA system and an Agilent 7500a ICP–MS equipped with a

shield torch. Analyses employed He as the ablation and carrier gas (0.65 L/min), a laser repetition rate of 6 Hz, and a laser spot size diameter of 32 μm. The analytical precision was better than 5% and GSE-1G was used as a standard. Silica contents determined by microprobe analysis were used to perform internal corrections for LA–ICP–MS data.

## 4. Results

### 4.1. Petrographic Description

Three Zuluhong biotite monzogranite samples were collected in the study area. The Zuluhong biotite monzogranite has a granitic and medium- to fine-grained texture and massive structure. The main rock-forming minerals are quartz (30–35 wt%), alkali feldspar (35–40 wt%), plagioclase (15–20 wt%), and biotite (~8%). No hornblende or muscovite are present. Accessory minerals (~2 wt%) include zircon, ilmenite, and apatite. Magmatic biotite (Figure 3a) in the Zuluhong monzogranite has an idiomorphic or hypidiomorphic texture and displays a distinct pleochroism (light to dark brown). Some accessory minerals (e.g., apatite and fluorite) are found as inclusions in the magmatic biotite. Around mineralized quartz veins, biotite is altered to chlorite. Where biotite is partially altered (Figure 3b,c), chloritization has generally occurred along cleavages and cracks, and the pleochroism is from light brown to brown. Some biotites are completely replaced by chlorite and have a pseudomorphic texture (Figure 3d). These biotites have an idiomorphic or hypidiomorphic texture and display a distinct pleochroism from light to dark green.

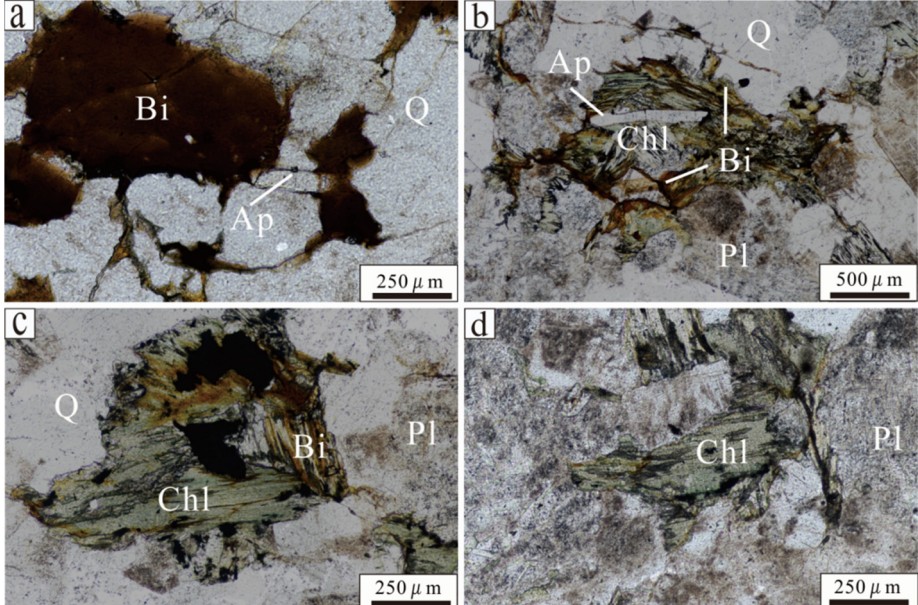

**Figure 3.** Photomicrographs of different biotite types from the Zuluhong monzogranite. (**a**) Unaltered biotite. (**b**,**c**) Partially altered biotite. (**d**) Biotite completely replaced by chlorite (Ap = apatite; Bi = biotite; Chl = chlorite; Pl = plagioclase; Q = quartz). All of the photomicrographs are in plane-polarized light.

### 4.2. Major Element Data

EMPA results for biotite and chlorite in the Zuluhong monzogranite are listed in Tables 1–3. The major element data for biotite were converted to a structural formula on the basis of 22 oxygens [30]. Data for chlorite were converted to a structural formula on the basis of 14 oxygens [31].

**Table 1.** Major (wt.%) element data of the unaltered and partially altered biotites in Zuluhong granite.

| Sample | U2911 | U2912 | U2921 | U2922 | U2923 | U2924 | U2925 | U2926 | U2927 | U29211 |
|---|---|---|---|---|---|---|---|---|---|---|
| Major element (wt.%) | | | | | | | | | | |
| $SiO_2$ | 35.60 | 35.52 | 34.88 | 34.92 | 35.50 | 34.44 | 34.44 | 35.27 | 34.50 | 35.96 |
| $TiO_2$ | 2.28 | 2.31 | 2.13 | 2.34 | 2.37 | 2.34 | 2.28 | 2.31 | 2.22 | 2.28 |
| $Al_2O_3$ | 15.28 | 15.28 | 14.92 | 15.02 | 15.07 | 14.68 | 15.00 | 14.63 | 14.39 | 15.58 |
| FeO | 25.16 | 24.97 | 25.25 | 25.53 | 25.38 | 25.24 | 25.75 | 25.49 | 24.92 | 25.77 |
| MnO | 0.49 | 0.49 | 0.54 | 0.48 | 0.50 | 0.52 | 0.52 | 0.57 | 0.48 | 0.47 |
| MgO | 6.86 | 6.92 | 7.03 | 6.89 | 6.59 | 6.90 | 6.72 | 6.90 | 7.06 | 6.57 |
| CaO | 0.09 | 0.04 | 0.02 | 0.04 | 0.03 | 0.09 | 0.06 | 0.10 | 0.07 | 0.01 |
| $Na_2O$ | 0.10 | 0.09 | 0.00 | 0.09 | 0.07 | 0.13 | 0.09 | 0.11 | 0.11 | 0.07 |
| $K_2O$ | 9.38 | 9.34 | 9.72 | 9.64 | 9.66 | 9.32 | 9.50 | 9.52 | 9.52 | 9.80 |
| Total | 95.24 | 94.96 | 94.54 | 94.95 | 95.16 | 93.65 | 94.37 | 94.90 | 93.27 | 96.51 |
| Number of ions on the basis of 22 oxygens | | | | | | | | | | |
| Si | 2.81 | 2.81 | 2.79 | 2.78 | 2.81 | 2.78 | 2.76 | 2.81 | 2.79 | 2.81 |
| $Al^{IV}$ | 1.19 | 1.19 | 1.21 | 1.22 | 1.19 | 1.22 | 1.24 | 1.19 | 1.21 | 1.19 |
| Ti | 0.14 | 0.14 | 0.13 | 0.14 | 0.14 | 0.14 | 0.14 | 0.14 | 0.14 | 0.13 |
| $Fe^{3+}$ | 0.17 | 0.18 | 0.12 | 0.13 | 0.16 | 0.13 | 0.12 | 0.14 | 0.12 | 0.16 |
| $Fe^{2+}$ | 1.49 | 1.47 | 1.57 | 1.57 | 1.52 | 1.57 | 1.61 | 1.56 | 1.57 | 1.52 |
| Mn | 0.03 | 0.03 | 0.04 | 0.03 | 0.03 | 0.04 | 0.04 | 0.04 | 0.03 | 0.03 |
| Mg | 0.81 | 0.81 | 0.84 | 0.82 | 0.78 | 0.83 | 0.80 | 0.82 | 0.85 | 0.76 |
| Ca | 0.01 | 0.00 | 0.00 | 0.00 | 0.00 | 0.01 | 0.01 | 0.01 | 0.01 | 0.00 |
| Na | 0.02 | 0.01 | 0.01 | 0.01 | 0.01 | 0.02 | 0.01 | 0.02 | 0.02 | 0.01 |
| K | 0.94 | 0.94 | 0.99 | 0.98 | 0.98 | 0.96 | 0.97 | 0.97 | 0.98 | 0.98 |
| Total | 7.83 | 7.82 | 7.88 | 7.87 | 7.84 | 7.87 | 7.88 | 7.86 | 7.88 | 7.84 |
| $Fe^{3+/2+}$ | 0.12 | 0.12 | 0.07 | 0.08 | 0.11 | 0.08 | 0.07 | 0.09 | 0.07 | 0.11 |
| Mg# | 0.53 | 0.53 | 0.53 | 0.52 | 0.51 | 0.53 | 0.52 | 0.52 | 0.54 | 0.51 |
| A/CNK | 1.46 | 1.48 | 1.41 | 1.41 | 1.40 | 1.42 | 1.37 | 1.35 | 1.49 | 2.45 |
| Buffer | NNO | NNO | NNO | NNO | NNO | NNO | NNO | NNO | NNO | NNO |

| Sample | P29210 | P2931 | P2932 | P2933 | P2934 | P2935 | P2936 | P2937 | P2938 | P2939 | P29310 |
|---|---|---|---|---|---|---|---|---|---|---|---|
| Major element (wt.%) | | | | | | | | | | | |
| $SiO_2$ | 25.13 | 23.45 | 25.40 | 23.19 | 24.78 | 24.77 | 24.42 | 24.29 | 33.16 | 25.19 | 25.61 |
| $TiO_2$ | 0.18 | 0.13 | 0.17 | 1.46 | 0.10 | 0.18 | 0.13 | 0.09 | 2.39 | 0.09 | 0.12 |
| $Al_2O_3$ | 20.98 | 19.58 | 17.79 | 18.95 | 20.24 | 19.97 | 20.58 | 20.43 | 13.68 | 20.40 | 20.34 |
| FeO | 33.59 | 32.98 | 31.87 | 32.92 | 32.96 | 33.10 | 32.78 | 32.87 | 22.55 | 34.77 | 34.48 |
| MnO | 0.72 | 0.56 | 0.37 | 0.59 | 0.60 | 0.48 | 0.63 | 0.67 | 0.51 | 1.13 | 0.98 |
| MgO | 8.64 | 9.69 | 10.01 | 8.91 | 8.90 | 9.23 | 9.13 | 8.62 | 6.25 | 7.35 | 7.55 |
| CaO | 0.01 | 0.04 | 0.07 | 0.03 | 0.02 | 0.02 | 0.01 | 0.01 | 0.00 | 0.00 | 0.00 |
| $Na_2O$ | 0.01 | 0.06 | 0.06 | 0.03 | 0.01 | 0.05 | 0.03 | 0.05 | 0.11 | 0.01 | 0.04 |
| $K_2O$ | 0.30 | 0.02 | 0.33 | 0.09 | 0.07 | 0.18 | 0.21 | 0.03 | 9.15 | 0.09 | 0.08 |
| Total | 89.56 | 86.51 | 86.07 | 86.18 | 87.68 | 87.98 | 87.91 | 87.03 | 87.80 | 89.04 | 89.19 |
| Number of ions on the basis of 22 oxygens | | | | | | | | | | | |
| Si | 2.24 | 2.07 | 2.15 | 2.14 | 2.11 | 2.12 | 2.84 | 2.17 | 2.19 | 2.14 | 2.08 |
| $Al^{IV}$ | 1.76 | 1.93 | 1.85 | 1.86 | 1.89 | 1.88 | 1.16 | 1.83 | 1.81 | 1.86 | 1.92 |
| $Al^{VI}$ | 0.09 | 0.05 | 0.22 | 0.18 | 0.21 | 0.23 | 0.21 | 0.24 | 0.25 | 0.24 | 0.12 |
| Ti | 0.01 | 0.10 | 0.01 | 0.01 | 0.01 | 0.01 | 0.15 | 0.01 | 0.01 | 0.01 | 0.01 |
| $Fe^{3+}$ | 0.15 | 0.15 | 0.18 | 0.16 | 0.16 | 0.18 | 0.17 | 0.21 | 0.22 | 0.18 | 0.10 |
| $Fe^{2+}$ | 2.20 | 2.30 | 2.21 | 2.24 | 2.21 | 2.23 | 1.44 | 2.30 | 2.25 | 2.21 | 2.34 |
| Mn | 0.03 | 0.04 | 0.04 | 0.04 | 0.05 | 0.05 | 0.04 | 0.08 | 0.07 | 0.05 | 0.04 |
| Mg | 1.32 | 1.18 | 1.15 | 1.19 | 1.18 | 1.12 | 0.80 | 0.94 | 0.96 | 1.09 | 1.28 |
| Ca | 0.01 | 0.00 | 0.00 | 0.00 | 0.00 | 0.00 | 0.00 | 0.00 | 0.00 | 0.00 | 0.00 |
| Na | 0.01 | 0.01 | 0.00 | 0.01 | 0.00 | 0.01 | 0.02 | 0.00 | 0.01 | 0.00 | 0.01 |
| K | 0.04 | 0.01 | 0.01 | 0.02 | 0.02 | 0.00 | 1.00 | 0.01 | 0.01 | 0.03 | 0.00 |
| Total | 7.85 | 7.85 | 7.82 | 7.84 | 7.84 | 7.82 | 7.83 | 7.79 | 7.78 | 7.82 | 7.90 |
| $Fe^{3+/2+}$ | 0.07 | 0.07 | 0.08 | 0.07 | 0.07 | 0.08 | 0.12 | 0.09 | 0.10 | 0.08 | 0.04 |
| Mg# | 0.51 | 0.55 | 0.56 | 0.53 | 0.52 | 0.53 | 0.53 | 0.52 | 0.53 | 0.46 | 0.47 |
| A/CNK | 56.94 | 97.27 | 31.24 | 94.18 | 156.41 | 63.94 | 71.31 | 167.43 | 1.35 | 163.88 | 133.84 |
| Buffer | FMQ | FMQ | FMQ | FMQ | FMQ | FMQ | FMQ | FMQ | FMQ | FMQ | FMQ |

U = Unaltered Biotite; P= Partially Altered Biotites; Mg# = molar Mg/(Mg + Fe); A/CNK = molar $Al_2O_3/(CaO + Na_2O + K_2O)$; $Fe^{3+/2+}$ = molar $Fe^{3+}/Fe^{2+}$; buffer = the oxygen buffer; "/" represents that the Ti content of partially altered biotites is below limit and the oxygen buffer cannot be estimated.

**Table 2.** LA-ICP-MS Analysis result data of magmatic biotites in the Zuluhong granite.

| Sample | Z331 | Z332 | Z333 | Z334 | Z335 | Z336 | Z3311 | Z3312 | Z3312 | Z3313 | Z3314 |
|---|---|---|---|---|---|---|---|---|---|---|---|
| | | | | | Trace element (ppm.) | | | | | | |
| Li | 638.14 | 440.45 | 451.24 | 536.22 | 693.73 | 333.63 | 595.63 | 563.42 | 587.53 | 540.43 | 600.71 |
| V | 238.17 | 355.5 | 309.87 | 295.94 | 286.91 | 185.76 | 267.59 | 311.62 | 289.39 | 288.97 | 253.91 |
| Cr | 32.35 | 29.65 | 28.88 | 53.35 | 38.28 | 26.17 | 47.68 | 60.98 | 56.90 | 55.12 | 35.81 |
| B | 1.01 | 2.86 | 4.76 | 1.21 | 2.16 | 0.00 | 0.09 | 0.00 | 1.41 | 0.24 | 2.26 |
| Sc | 64.43 | 74.59 | 60.36 | 63.13 | 60.55 | 33.21 | 67.89 | 69.86 | 69.18 | 70.75 | 65.43 |
| Co | 33.08 | 43.30 | 40.93 | 38.49 | 33.39 | 27.60 | 35.32 | 37.10 | 37.41 | 36.32 | 34.60 |
| Ni | 20.90 | 25.66 | 26.11 | 22.31 | 15.39 | 18.22 | 26.92 | 26.77 | 26.88 | 27.08 | 23.35 |
| Cu | 1.51 | 7.62 | 5.85 | 12.50 | 1.40 | 6.43 | 2.89 | 0.36 | 1.24 | 0.29 | 0.60 |
| Zn | 716.65 | 1005.35 | 1001.45 | 859.73 | 719.19 | 604.10 | 653.67 | 624.31 | 673.23 | 720.86 | 769.59 |
| Ga | 65.37 | 89.42 | 84.80 | 74.68 | 62.96 | 55.68 | 58.87 | 60.79 | 63.49 | 62.84 | 61.49 |
| Rb | 977.18 | 241.28 | 437.40 | 779.90 | 1094.26 | 1067.27 | 542.97 | 611.72 | 637.47 | 762.52 | 794.84 |
| Sr | 1.38 | 2.47 | 2.09 | 7.14 | 1.08 | 4.61 | 754.02 | 8.76 | 0.88 | 0.34 | 0.97 |
| Y | 0.59 | 1.37 | 0.59 | 6.29 | 0.63 | 1.77 | 2.51 | 0.33 | 0.52 | 0.05 | 1.19 |
| Zr | 0.45 | 0.46 | 0.72 | 2.22 | 0.40 | 23.03 | 0.66 | 0.18 | 2.18 | 0.08 | 0.23 |
| Nb | 122.38 | 28.14 | 39.63 | 96.00 | 136.04 | 70.30 | 78.82 | 82.69 | 89.38 | 106.54 | 130.31 |
| Mo | 0.09 | 0.00 | 0.28 | 0.42 | 0.00 | 0.35 | 0.56 | 0.34 | 0.15 | 0.24 | 0.09 |
| Sn | 74.16 | 17.32 | 24.09 | 44.22 | 74.93 | 80.44 | 66.50 | 60.92 | 66.28 | 69.10 | 79.93 |
| Cs | 52.15 | 28.81 | 105.28 | 68.62 | 57.57 | 167.77 | 29.19 | 18.96 | 15.78 | 20.68 | 27.04 |
| **Sample** | **Z331** | **Z332** | **Z333** | **Z334** | **Z335** | **Z336** | **Z3311** | **Z3312** | **Z3312** | **Z3313** | **Z3314** |
| | | | | | Trace element (ppm.) | | | | | | |
| Ba | 1130.20 | 229.05 | 34.87 | 80.07 | 894.13 | 140.05 | 2000.64 | 575.74 | 677.56 | 397.51 | 1500.49 |
| La | 0.17 | 0.35 | 0.39 | 1.42 | 0.11 | 1.61 | 0.86 | 0.05 | 0.11 | 0.02 | 0.17 |
| Ce | 0.17 | 0.64 | 0.60 | 1.99 | 0.26 | 3.34 | 0.23 | 0.03 | 0.16 | 0.01 | 0.14 |
| Pr | 0.06 | 0.13 | 0.11 | 0.72 | 0.02 | 0.74 | 0.24 | 0.01 | 0.05 | 0.00 | 0.05 |
| Nd | 0.25 | 0.83 | 0.55 | 2.7 | 0.28 | 2.23 | 1.26 | 0.07 | 0.14 | 0.03 | 0.24 |
| Sm | 0.08 | 0.18 | 0.17 | 0.98 | 0.08 | 0.79 | 0.25 | 0.01 | 0.06 | 0.00 | 0.22 |
| Eu | 0.00 | 0.02 | 0.03 | 0.04 | 0.02 | 0.14 | 0.06 | 0.02 | 0.00 | 0.02 | 0.06 |
| Gd | 0.05 | 0.30 | 0.08 | 1.11 | 0.04 | 0.31 | 0.36 | 0.00 | 0.01 | 0.00 | 0.20 |
| Tb | 0.01 | 0.02 | 0.01 | 0.22 | 0.04 | 0.06 | 0.03 | 0.00 | 0.01 | 0.00 | 0.02 |
| Dy | 0.11 | 0.24 | 0.15 | 1.30 | 0.09 | 0.46 | 0.35 | 0.06 | 0.06 | 0.02 | 0.14 |
| Ho | 0.01 | 0.03 | 0.02 | 0.25 | 0.04 | 0.11 | 0.06 | 0.01 | 0.02 | 0.00 | 0.05 |
| Er | 0.08 | 0.17 | 0.07 | 0.44 | 0.04 | 0.04 | 0.35 | 0.04 | 0.07 | 0.00 | 0.12 |
| Tm | 0.01 | 0.01 | 0.00 | 0.08 | 0.03 | 0.00 | 0.03 | 0.01 | 0.01 | 0.00 | 0.02 |
| Yb | 0.04 | 0.19 | 0.141 | 0.60 | 0.08 | 0.23 | 0.24 | 0.03 | 0.09 | 0.02 | 0.14 |
| Lu | 0.01 | 0.01 | 0.02 | 0.07 | 0.03 | 0.01 | 0.02 | 0.00 | 0.01 | 0.01 | 0.01 |
| Hf | 0.03 | 0.04 | 0.07 | 0.06 | 0.06 | 1.33 | 0.04 | 0.00 | 0.24 | 0.01 | 0.04 |
| Ta | 22.06 | 5.08 | 7.96 | 18.76 | 25.96 | 10.90 | 5.95 | 5.11 | 6.36 | 11.18 | 28.72 |
| W | 0.97 | 0.34 | 0.61 | 1.09 | 0.97 | 0.65 | 0.42 | 0.61 | 0.75 | 0.99 | 0.86 |
| Pb | 3.64 | 10.79 | 12.48 | 17.19 | 3.17 | 7.72 | 6.07 | 2.76 | 3.72 | 2.28 | 3.33 |
| Th | 0.07 | 0.64 | 1.23 | 2.13 | 0.00 | 2.28 | 0.03 | 0.02 | 0.14 | 0.00 | 0.01 |
| U | 0.02 | 0.16 | 0.27 | 0.88 | 0.00 | 0.62 | 0.03 | 0.00 | 0.03 | 0.01 | 0.01 |
| ΣREE | 1.04 | 3.12 | 2.32 | 11.93 | 1.17 | 10.08 | 4.35 | 0.357 | 0.80 | 0.13 | 1.57 |
| LREE | 0.72 | 2.14 | 1.84 | 7.85 | 0.77 | 8.86 | 2.90 | 0.19 | 0.53 | 0.08 | 0.87 |
| HREE | 0.32 | 0.98 | 0.49 | 4.08 | 0.39 | 1.22 | 1.45 | 0.16 | 0.28 | 0.05 | 0.70 |
| L/H | 2.25 | 2.20 | 3.75 | 1.93 | 1.93 | 7.24 | 1.99 | 1.20 | 1.90 | 1.68 | 1.26 |
| L/Y | 2.71 | 1.34 | 1.96 | 1.69 | 1.04 | 4.98 | 2.50 | 0.97 | 0.95 | 0.49 | 0.87 |
| δEu | 0.93 | 0.24 | 0.78 | 0.11 | 0.99 | 0.72 | 0.62 | 4.04 | 0.58 | 4.46 | 0.92 |
| δCe | 0.43 | 0.73 | 0.71 | 0.48 | 1.17 | 0.75 | 0.12 | 0.39 | 0.50 | 0.39 | 0.36 |

L/H = LREE/HREE; L/Y = LaN/YbN.

**Table 3.** Major (wt.%) element data of the chlorites in Zuluhong granite (during ore formation).

| Sample | C1 | C3 | C4 | C5 | C6 | C7 | C8 | C9 | C10 | C11 | C12 | C13 | C15 |
|---|---|---|---|---|---|---|---|---|---|---|---|---|---|
| | | | | | Major element (wt.%) | | | | | | | | |
| $SiO_2$ | 25.95 | 26.36 | 24.99 | 24.92 | 25.15 | 25.10 | 25.09 | 25.06 | 24.73 | 24.33 | 24.55 | 23.92 | 23.81 |
| $TiO_2$ | 0.02 | 0.05 | 0.02 | 0.03 | 0.06 | 0.08 | 0.00 | 0.00 | 0.01 | 0.00 | 0.02 | 0.01 | 0.11 |
| $Al_2O_3$ | 15.48 | 15.86 | 20.19 | 20.35 | 20.13 | 19.82 | 20.99 | 20.37 | 20.32 | 20.06 | 19.71 | 19.38 | 19.23 |
| FeO | 36.37 | 32.98 | 29.82 | 29.32 | 29.39 | 29.70 | 29.75 | 30.39 | 30.78 | 29.70 | 30.21 | 29.31 | 29.46 |
| MnO | 0.65 | 0.56 | 0.53 | 0.56 | 0.56 | 0.53 | 0.61 | 0.66 | 0.62 | 0.60 | 0.63 | 0.60 | 0.63 |
| MgO | 6.71 | 8.86 | 9.96 | 9.54 | 9.33 | 9.86 | 9.21 | 9.41 | 9.39 | 9.91 | 9.42 | 9.91 | 10.43 |
| CaO | 0.00 | 0.03 | 0.02 | 0.03 | 0.06 | 0.08 | 0.09 | 0.04 | 0.07 | 0.07 | 0.07 | 0.10 | 0.15 |
| $Na_2O$ | 0.05 | 0.07 | 0.06 | 0.12 | 0.12 | 0.12 | 0.13 | 0.05 | 0.05 | 0.04 | 0.04 | 0.06 | 0.09 |
| $K_2O$ | 0.01 | 0.05 | 0.05 | 0.10 | 0.06 | 0.06 | 0.04 | 0.01 | 0.04 | 0.00 | 0.01 | 0.04 | 0.02 |
| $Cr_2O_3$ | 0.02 | 0.03 | 0.01 | 0.06 | 0.04 | 0.05 | 0.14 | 0.00 | 0.02 | 0.05 | 0.05 | 0.28 | 0.08 |
| Total | 85.26 | 84.85 | 85.64 | 85.04 | 84.89 | 85.41 | 86.05 | 85.97 | 86.03 | 84.74 | 84.70 | 83.61 | 84.00 |
| | | | | | Number of ions on the basis of 14 oxygens | | | | | | | | |
| Si | 3.02 | 3.02 | 2.78 | 2.78 | 2.81 | 2.80 | 2.77 | 2.78 | 2.75 | 2.74 | 2.77 | 2.74 | 2.71 |
| $Al^{IV}$ | 0.98 | 0.98 | 1.22 | 1.22 | 1.19 | 1.20 | 1.23 | 1.22 | 1.25 | 1.26 | 1.23 | 1.26 | 1.29 |
| $Al^{VI}$ | 1.14 | 1.16 | 1.42 | 1.46 | 1.47 | 1.40 | 1.50 | 1.44 | 1.42 | 1.40 | 1.39 | 1.35 | 1.30 |
| Ti | 0.002 | 0.004 | 0.002 | 0.003 | 0.005 | 0.007 | 0.00 | 0.00 | 0.001 | 0.00 | 0.002 | 0.001 | 0.009 |
| Cr | 0.002 | 0.03 | 0.001 | 0.005 | 0.004 | 0.004 | 0.012 | 0.00 | 0.002 | 0.004 | 0.005 | 0.025 | 0.007 |
| $Fe^{2+}$ | 3.54 | 3.16 | 2.77 | 2.74 | 2.75 | 2.77 | 2.75 | 2.82 | 2.86 | 2.79 | 2.85 | 2.80 | 2.81 |
| Mn | 0.06 | 0.05 | 0.05 | 0.05 | 0.05 | 0.05 | 0.06 | 0.06 | 0.06 | 0.06 | 0.06 | 0.06 | 0.06 |
| Mg | 1.16 | 1.51 | 1.65 | 1.59 | 1.56 | 1.64 | 1.52 | 1.56 | 1.56 | 1.66 | 1.59 | 1.69 | 1.77 |
| Ca | 0.000 | 0.004 | 0.002 | 0.004 | 0.007 | 0.010 | 0.011 | 0.005 | 0.008 | 0.008 | 0.008 | 0.012 | 0.018 |
| Na | 0.011 | 0.015 | 0.013 | 0.026 | 0.026 | 0.026 | 0.028 | 0.011 | 0.011 | 0.009 | 0.009 | 0.013 | 0.020 |
| K | 0.002 | 0.007 | 0.007 | 0.014 | 0.009 | 0.009 | 0.006 | 0.001 | 0.006 | 0.000 | 0.001 | 0.006 | 0.003 |
| Total | 9.92 | 9.92 | 9.91 | 9.90 | 9.87 | 9.91 | 9.88 | 9.90 | 9.92 | 9.93 | 9.92 | 9.95 | 9.99 |
| Mg# | 0.43 | 0.52 | 0.58 | 0.57 | 0.56 | 0.56 | 0.56 | 0.55 | 0.58 | 0.56 | 0.58 | 0.59 | 0.56 |
| $lg(fO_2)$ | −43.58 | −44.08 | −39.93 | −40.05 | −40.57 | −39.84 | −39.93 | −39.42 | −39.33 | −39.78 | −39.25 | −38.85 | −40.48 |
| T(°C) | 230.82 | 222.31 | 264.64 | 239.62 | 224.75 | 246.69 | 231.12 | 253.31 | 270.39 | 282.87 | 264.92 | 278.56 | 302.34 |

C = Chlorite; Mg# = molar Mg/(Mg + Fe); T(°C) = Centigrade temperature.

When data for magmatic and partially altered biotites are plotted on the $Mg^-(Al^{3+} + Fe^{3+} + Ti^{4+})-(Fe^{2+} + Mn^{2+})$ diagram [32], they all fall in the Fe-biotite field (Figure 4a). On the classification diagram for chlorite [33], the data fall in the field for ripidolite (Figure 4a). Biotite and chlorite are both Fe-rich and show a distinct progression in chemistry from biotite to chlorite. According to Inoue [34], Fe-rich chlorites tend to form in a reducing environment. As such, the Fe-rich chlorite in the Zuluhong monzogranite may indicate a low oxygen fugacity during and after alteration.

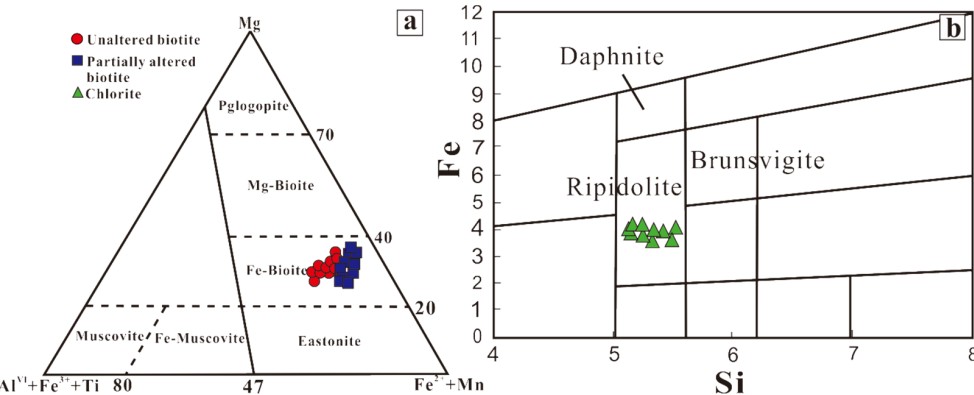

**Figure 4.** Compositional plots for biotite and chlorite. (**a**) Compositional plot for biotite, showing that the data fall in the Fe-biotite field (modified after Foster, [32]). (**b**) Compositional plot for chlorite, showing that the data fall in the ripidolite field (modified after Deer, [33]).

The chemical changes that accompanied biotite alteration were as follows. (1) Biotite alteration results in a decrease in Ti, K, and Si contents. For example, the $TiO_2$ contents from magmatic biotite to partially altered biotite and chlorite are 1.95–2.37, 0.01–2.34, and 0–0.10 wt.%, respectively, indicating the release of Ti from biotite during alteration. The $K_2O$ contents of biotite, partially altered biotite, and chlorite are 8.67–9.72, 0.18–0.31, and 0–0.05 wt.%, and the $SiO_2$ contents are 33.80–38.09, 23.5–33.2, and 24.24–25.95 wt.%, respectively. The gradual loss of K and Si from the biotite provides lattice space for the entry of other elements. (2) Fe, Mg, and Al contents show an increasing trend from magmatic biotite to partially altered biotite and chlorite, with $FeO_{tot}$ contents of 24.6–26.1 wt.% ($Fe^{3+}$ = 0.1–0.22 wt.%; $Fe^{2+}$ = 1.50–2.32 wt.%), 31.90–33.60 wt.% ($Fe^{3+}$ = 0.1–0.21 wt.%; $Fe^{2+}$ = 2.20–2.34 wt.%), and 29.32–36.37 wt.% ($Fe^{3+}$ = 0.09–0.25 wt.%; $Fe^{2+}$ = 5.50–6.91 wt.%), respectively. Similarly, the MgO contents are 6.6–7.8, 8.64–10.01, and 6.71–10.42 wt.%, and the $Al_2O_3$ contents are 14.6–15.3, 17.8–20.9, and 15.5–20.9 wt.%, respectively. The systematic variations in Fe, Mg, and Al contents reflect changes in the biotite crystalline structure during alteration. In addition, the changing $FeO_{tot}$ and, in particular, the decreasing $Fe^{3+}/Fe^{2+}$ ratio, indicate a progressive decrease in oxygen fugacity. (3) Mn, Ca, and Na contents show no significant variation, indicating that they were largely unaffected by alteration and physicochemical and redox conditions.

### 4.3. Rare Earth Element Data

Previous studies have indicated that biotite has low REE contents. For example, Pan et al. [35] reported ∑REE contents of 2.54–5.62 ppm for biotite from the northwestern Yunnan porphyry Cu belt, and Bea et al. [36] determined ∑REE biotite contents of 0.28–1.44 ppm for the Pena Negra complex, central Spain.

In this study, REE contents in biotite are also relatively low. Biotite in the Zuluhong monzogranite has ∑REE contents of 0.36–11.93 ppm (average = 3.40 ppm), light REE contents of 0.19–8.86 ppm (average = 2.45 ppm), heavy REE contents of 0.05–4.08 ppm (average = 0.95 ppm), and light REE/heavy REE ratios of 1.26–7.24 (average = 2.34), reflecting slight depletion of the heavy REE. The chondrite-normalized REE patterns for the Zuluhong biotites (Figure 5a) are relative flat, with a slight negative Ce anomaly. In addition, whole-rock ∑REE contents of the Zuluhong monzogranite are 127–154 ppm, and the ∑REE contents of the biotite account for 5–10% of the whole-rock REE inventory.

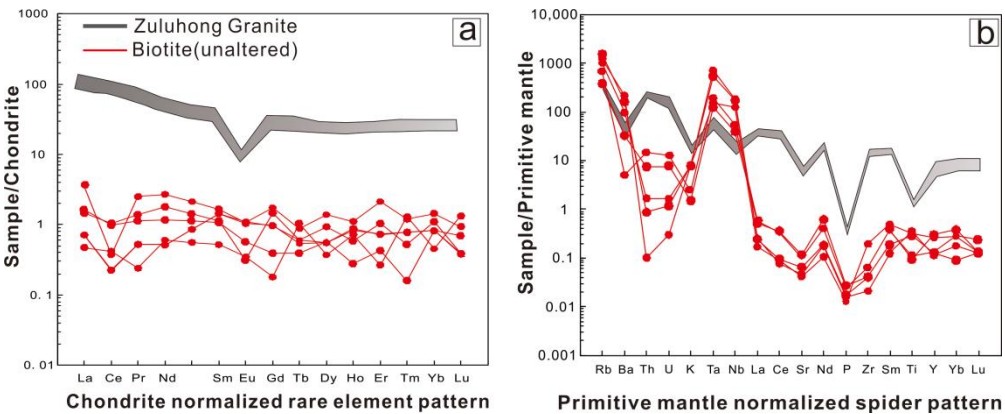

**Figure 5.** Rare earth and trace element patterns of the Zuluhong monzogranite and biotite. (**a**) Chondrite-normalized rare earth element patterns. (**b**) Primitive-mantle-normalized trace element patterns [28].

### 4.4. Trace Element Data

The Zuluhong monzogranite is enriched in large-ion lithophile elements (LILE; Rb and Sr) and depleted in high-field-strength elements (HFSE; Nb and Ti). Compared with Rb and Th, the monzogranite is depleted in Ba. The fact that the Zuluhong monzogranite is depleted in HFSE (e.g., Nb and Ta) and enriched in Zr and Sm indicates the granitic magma

was formed by crustal melting (Figure 5b) [28]. Compared with the granite, the biotites are enriched in LILE (e.g., Rb, Sr, Ba, Na, and U) and depleted in HFSE (e.g., Nb, Ta, Zr, Hf, and Th), showing that biotite was the main carrier of some of these elements (Rb, Ba, Ta, Nb, and K) in the monzogranitic magma.

## 5. Discussion

### 5.1. Temperature and Oxygen Fugacity Conditions

Experimental studies that determined oxygen fugacity from biotite coexisting with magnetite and K-feldspar have used $Fe^{3+}$, $Fe^{2+}$, and $Mg^{2+}$ contents [37]. In a $Fe^{3+}$–$Fe^{2+}$–$Mg^{2+}$ diagram (Figure 6), data for biotites plot on the Ni–NiO buffer line. In conjunction with the formation temperature, the $\log(fO_2)$ value has an estimated range from −23 to −27 (average = −23.4). Data for the partially altered biotites plot on the FMQ buffer line, indicating a decrease in temperature and oxygen fugacity as the biotite was altered.

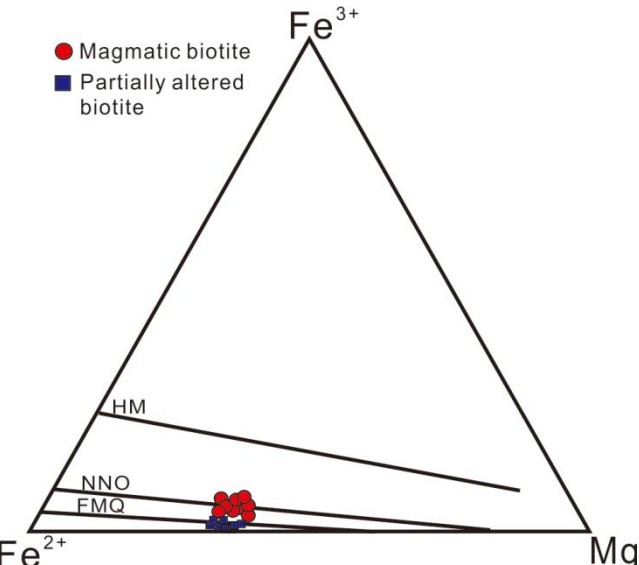

**Figure 6.** $Fe^{3+}$–$Fe^{2+}$–Mg diagram for biotite (modified after Wones et al. [37]).

Temperature and oxygen fugacity have a strong influence on the crystalline structure and chemical composition of chlorite [38–41]. Cathelineau and Nieva [38] first reported a positive correlation between the $Al^{VI}$ content of chlorite and formation temperature, and proposed a solid solution thermometer for chlorite. Nieto [39] reported a modified formula for the crystalline interplanar spacing ($d_{001}$) of chlorite: $d_{001} = 14.339 − 0.1155Al^{IV} − 0.02011 × Fe^{2+}$. Furthermore, a mathematical relationship between $d_{001}$ and the formation temperature of chlorite was proposed by Battaglia [40]: $T = (14.379 − d_{001})/0.001$. However, the reliability of this method is not ideal because the influence of whole-rock composition is not considered. Moreover, the chlorite formation temperature can also be calculated by using these equations [42,43]. Combined with the semi-empirical thermometer of Walshe [44], Inoue [41] has selected four chlorite components to establish a new thermodynamic model. Therefore, according to the method of Inoue, the formation temperature of chlorite in this study is 222.31–302.34 °C (average = 254.8 °C).

In view of the Fe/(Fe + Mg) ratio of chlorite (>0.6), the formula of Walshe [44] was used to estimate oxygen fugacity, yielding $\log(fO_2)$ values from −38.85 to −44.08 (average = −40.39).

### 5.2. Mechanisms and Physicochemical Conditions of Alteration

The general chemical formula of biotite is $XY_{2–3}[Z_4O_{10}](OH)_2$ (X mainly refers to $K^+$ and $Na^+$, as well as some $Ca^{2+}$, $Ba^{2+}$, $Rb^+$, or $Cs^+$; Y mainly refers to $Mg^{2+}$, $Fe^{2+}$, $Mn^{2+}$, $Fe^{3+}$, $Al^{3+}$, or $Ti^{4+}$; $OH^−$ can be replaced by $Li^+$, $F^−$, or $Cl^−$; Z mainly refers to $Si^{4+}$ or $Al^{3+}$

in the tetrahedral coordination structure) [45]. The crystalline structure of biotite is the TMT+C type (the T layer is the silica tetrahedron and corresponds to Z; the M layer has an octahedral structure and corresponds to Y; the C layer refers to $K^+$ and corresponds to X). The alteration of biotite commonly results in the formation of chlorite, which has a chemical formula of $Y_3[Z_4O_{10}](OH)_2$–$Y_3(OH)_6$ (Y mainly refers to $Mg^{2+}$, $Fe^{2+}$, $Mn^{2+}$, $Fe^{3+}$, $Al^{3+}$, or $Ti^{4+}$; $OH^-$ can be replaced by $Li^+$, $F^-$, or $Cl^-$). The crystalline structure of chlorite is the TMT+M type. The chemical formulae indicate that with the alteration of biotite, X ($K^+$ or $Na^+$) decreases but Y ($Fe^{2+}$ and $Al^{VI}$) increases. In addition, the C layer is gradually replaced by the M layer during alteration, and the C layer ($K^+$) eventually completely disappears; furthermore, increasing $Fe^{2+}$ and $Mg^{2+}$ may form an extra M layer, replacing the C layer [46].

Changes in physicochemical conditions during alteration can be reflected by mineral chemistry. The chloritization of biotite is a complex water-rock reaction process, which is controlled by temperature, oxygen fugacity, water/rock ratio, and fluid composition. It is generally accepted that the contents of Fe, Si, Ti, K, Al, and Mg recorded the greatest changes during the alteration of biotite. Previous studies have suggested that temperature shows a positive correlation with Ti content [37]. Some elements (such as Mg and Al) in the M layer of biotite can be replaced and substituted by Ti. Therefore, a continuous decline in Ti content during alteration might reflect a gradual decrease in temperature. Changes in K, Si, and $Al^{VI}$ contents are related to variations in crystalline structure. However, the contents of Fe, $Al^{VI}$, and Mg are mainly related to changes in oxygen fugacity. Higher $Fe^{3+}/Fe^{2+}$ ratios reflect a higher oxygen fugacity. In Figure 7, $Fe^{3+}/Fe^{2+}$ ratios negatively correlate with FeO, and $Al_2O_3$ and MgO contents positively correlate with $TiO_2$, indicating these changes reflect an increase in oxygen fugacity. This result is consistent with the negative correlation between $Al^{VI}$ in biotite and oxygen fugacity, as reported by Albuquerque [47].

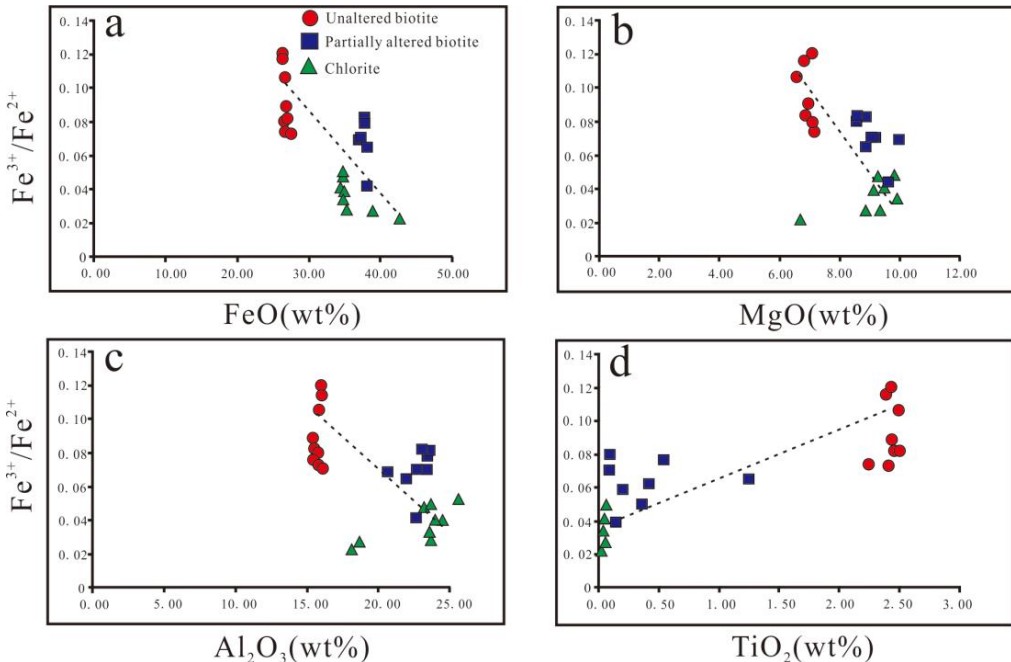

**Figure 7.** Biotite $Fe^{3+}/Fe^{2+}$ ratio versus major element oxides. (**a**) the negative relationships between $Fe^{3+}/Fe^{2+}$ and FeO. (**b**) the negative relationships between $Fe^{3+}/Fe^{2+}$ and MgO. (**c**) the negative relationships between $Fe^{3+}/Fe^{2+}$ and $Al_2O_3$. (**d**) the positive relationship between $Fe^{3+}/Fe^{2+}$ and $TiO_2$.

### 5.3. Petrogenesis

5.3.1. Magmatic Differentiation

Previous studies have suggested that biotite is a good indicator of magmatic differentiation [48]. Rb and Ba in granitic magmas are mainly substituted into biotite and K-feldspar.

Rb mainly substitutes for K, and increasing Rb content reflects magmatic differentiation. Ba can substitute for K and Ca, and during the early stages of magmatic evolution, biotite is the main host of Ba, whereas during the later stages, Ba is mainly substituted into K-feldspar. As such, magmatic evolution results in a gradual increase of Rb and decrease of Ba in biotite, and an increase in the Rb/Ba ratio. The Zuluhong monzogranite has high Rb/Ba ratios (0.27–9.14; average = 2.97), clearly indicating that it is highly differentiated [28]. This is consistent with other mineralogical and geochemical characteristics, such as its weakly peraluminous nature, absence of hornblende, high $SiO_2$ (average 73.66 wt.%), high alkalis (average $Na_2O + K_2O$ = 7.45 wt.%), high K/Ba ratio (average = 130), and depletions in Eu, Sr, Ba, and Nb.

Previous studies have suggested that biotite records systematic chemical variation during magmatic evolution [49,50]. Under the microscopic observation, we found that biotite had different optical characteristics, which may be caused by the differences in composition of a zoned biotite (Figure 8a). Thus, major element composition analysis from point 1 to point 5 of a single biotite grain was carried out by the electron microprobe analyzer (Figure 8b). The MgO content increases gradually from point 1 to point 5 (Figure 9a), whereas the $FeO_{tot}$ content decreases (Figure 9b). The coupled decreases in MgO and $TiO_2$ during magma evolution (Figure 9c) show that the temperature decreased during biotite crystallization and that the changes in biotite and magmatic Fe and Mg contents reflect the fractional crystallization of biotite. During the evolution of the magma, the $Na_2O + K_2O$ content increased (Figure 9d), indicating that the magma became alkali rich. $Al_2O_3$ and CaO decreased during magma evolution (Figure 9e,f), possibly caused by the fractional crystallization of feldspar (Al-rich) and apatite (Ca-rich).

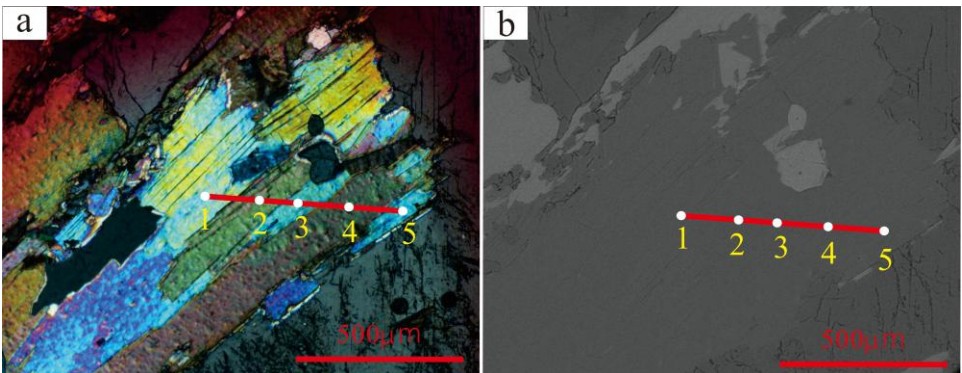

**Figure 8.** (**a**) Microphotograph of the biotite. (**b**) EDX map of the biotite, which exhibits heterogeneous composition.

5.3.2. Granite Petrogenesis

The biotite in the Zuluhong granite is Fe-rich and the main opaque mineral is ilmenite, indicating that the granite is an ilmenite-series granite [51]. The distribution of Fe and Mg in biotite is controlled mainly by the Mg/Fe ratio in the magma and temperature. In this study, Mg# (molar Mg/(Mg + Fe)) is used to represent the Mg/Fe ratio. The Fe and Mg contents of biotite show different trends with temperature, i.e., as temperature increases, the biotite becomes Mg-rich. The Mg# value of biotite in the Zuluhong granite is 0.45–0.55, which indicates a low crystallization temperature. Magnetite-series granitoids are considered to have deep sources (upper mantle or lowest crust) and contain Mg-rich biotite, whereas ilmenite-series granitoids have relatively shallow sources (middle to lower continental crust) and contain Fe-rich biotite. The MF value (molar Mg/(Mg + Fe + Mn)) of biotite is also an important indicator in identifying the magma source. The MF value of biotite in mantle rocks is >0.45, whereas in crustal rocks it is <0.45. The MF values of biotite in the Zuluhong granite are 0.28–0.35 and the granite is Fe-rich, indicating that it was derived from the lower crust [52].

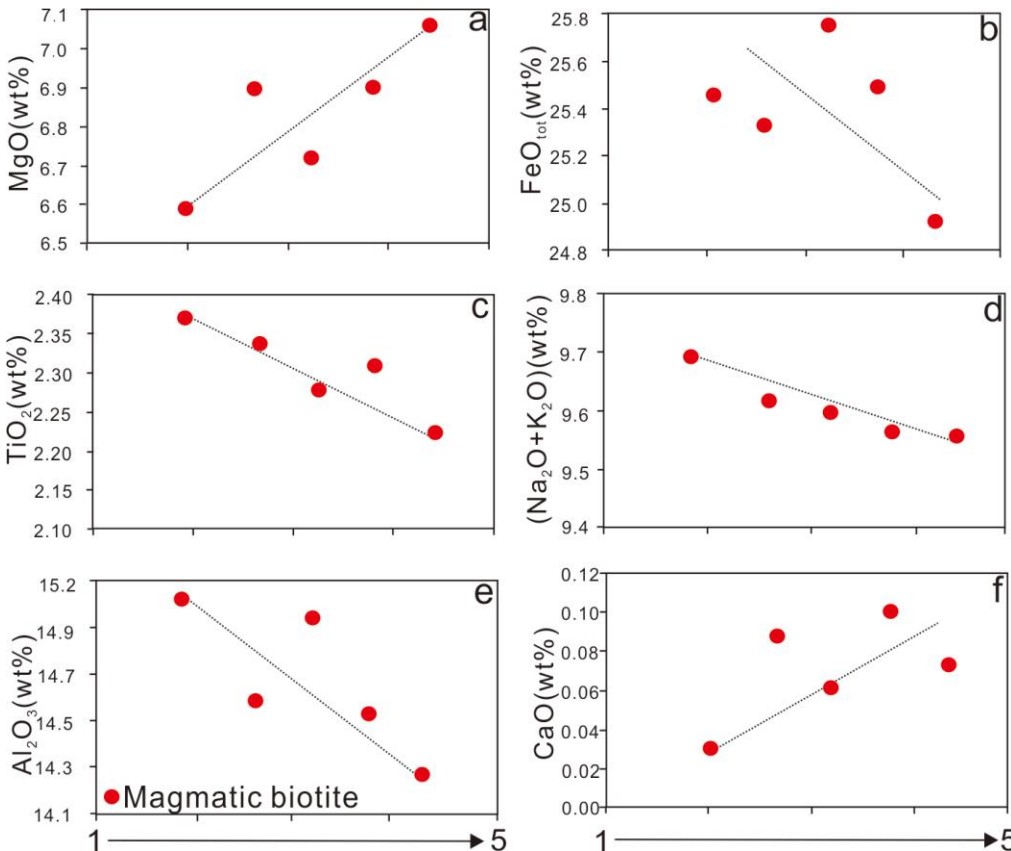

**Figure 9.** Major element variation diagrams for igneous biotite. (**a**) The MgO content of igneous biotite increases from the Point 1 to Point 5. (**b**) The FeO content of igneous biotite decreases from the Point 1 to Point 5. (**c**) The TiO$_2$ content of igneous biotite increases from the Point 1 to Point 5. (**d**) The (Na$_2$O + K$_2$O) content of igneous biotite increases from the Point 1 to Point 5. (**e**) The Al$_2$O$_3$ content of igneous biotite increases from the Point 1 to Point 5. (**f**) The CaO content of igneous biotite increases from the Point 1 to Point 5. 1 = Point 1 of Figure 8 and 5 = Point 5 of Figure 8.

Aluminum saturation in granites is key to distinguishing between I- and S-type granites, with the differences between these granites reflecting the source rock and fractional crystallization processes. In this study, the A/CNK value (molar Al$_2$O$_3$/(Na$_2$O + K$_2$O + CaO)) is used to represent the degree of aluminum saturation. Major (wt.%) and trace ($\times 10^{-6}$) element data of the Zuluhong granite are listed in Table 4 [28]. The source rocks of S-type granites are derived from the middle–upper crust and are typically highly weathered sedimentary rocks in which Al is relatively stable, but Na, K, and Ca have been lost due to weathering, thereby producing peraluminous granites. I-type granite source rocks are undersaturated or oversaturated in Al and, as in the Zuluhong granite, fractional crystallization changes the Al content in the evolving magma. Zen [53] noted that hornblende crystallization (A/CNK < 0.5) is inevitable in the production of peraluminous melts. According to the Bowen reaction sequence, amphibole crystallizes before biotite. Given that there is no hornblende in the Zuluhong granite, it is clear that the granite is highly differentiated. Na, K, Ca, and Mg are removed from melt by the fractional crystallization of minerals with low A/CNK values (e.g., pyroxene and hornblende), which results in the formation of peraluminous melt. Chappell [54] proposed that the generation of weakly peraluminous granites is controlled mainly by the nature of the source rock, rather than fractional crystallization, and so S-type granites are always more Al-oversaturated than I-type granites. Gao [55] developed a Mg# versus A/CNK classification diagram for biotite and, in this diagram, data for the Zuluhong granite plot in the I-type granite field (Figure 10). This is consistent with the absence of the diagnostic primary Al-rich minerals (e.g., muscovite and cordierite) of S-type granites in the Zuluhong granite. As such, the Al-saturation in

the Zuluhong granite must be the result of fractional crystallization. In addition, the $Al^{VI}$ value of biotite in the Zuluhong granite is 0.17–0.26, which is characteristic of the lower $Al^{VI}$ values of biotite in I-type granites.

**Table 4.** Major (wt.%) and trace ($\times 10^{-6}$) element data of the Zuluhong granite.

| Simple | ZLH-29 | ZLH-30 | ZLH-31 | ZLH-33 | ZLH-34 | ZLH-26 | ZLH-27 | ZLH-1 | ZLH-2 |
|---|---|---|---|---|---|---|---|---|---|
| | | | | Major element (wt.%) | | | | | |
| $SiO_2$ | 72.70 | 73.22 | 74.03 | 74.41 | 73.77 | 63.72 | 64.12 | 65.30 | 64.38 |
| $TiO_2$ | 0.33 | 0.28 | 0.29 | 0.26 | 0.30 | 0.61 | 0.61 | 0.58 | 0.60 |
| $Al_2O_3$ | 13.44 | 13.53 | 13.18 | 13.14 | 13.62 | 15.98 | 15.99 | 16.01 | 15.99 |
| MnO | 0.07 | 0.06 | 0.06 | 0.06 | 0.06 | 0.10 | 0.10 | 0.10 | 0.10 |
| MgO | 0.62 | 0.51 | 0.56 | 0.49 | 0.59 | 2.49 | 2.53 | 2.27 | 2.43 |
| CaO | 1.74 | 1.60 | 1.59 | 1.42 | 1.69 | 4.41 | 4.61 | 4.03 | 4.35 |
| $Na_2O$ | 3.62 | 3.55 | 3.37 | 3.45 | 3.61 | 3.83 | 3.78 | 3.72 | 3.78 |
| $K_2O$ | 3.68 | 3.99 | 4.00 | 3.94 | 3.82 | 2.26 | 2.30 | 2.60 | 2.39 |
| $P_2O_5$ | 0.10 | 0.09 | 0.09 | 0.07 | 0.09 | 0.20 | 0.20 | 0.18 | 0.19 |
| $TFe_2O_3$ | 2.54 | 2.10 | 2.25 | 2.13 | 2.31 | 4.19 | 4.43 | 3.38 | 4.00 |
| LOI | 0.54 | 0.36 | 0.44 | 0.49 | 0.46 | 1.38 | 1.14 | 1.41 | 1.31 |
| Mg# | 32 | 32 | 33 | 33 | 32 | 54 | 53 | 57 | 54 |
| A/CNK | 1.03 | 1.04 | 1.03 | 1.05 | 1.04 | 0.95 | 0.94 | 0.98 | 0.96 |
| $Na_2O/K_2O$ | 0.98 | 0.89 | 0.84 | 0.88 | 0.95 | 1.69 | 1.64 | 1.43 | 1.58 |
| | | | | Trace elements ($10^{-6}$) | | | | | |
| La | 26.50 | 28.20 | 22.7 | 22.00 | 27.40 | 19.40 | 18.50 | 21.30 | 19.73 |
| Ce | 66.40 | 63.70 | 58.1 | 55.80 | 62.30 | 44.70 | 41.10 | 46.30 | 44.0 |
| Pr | 6.90 | 7.27 | 6.00 | 6.00 | 7.14 | 5.30 | 5.04 | 5.60 | 5.31 |
| Nd | 26.10 | 27.30 | 22.6 | 22.70 | 26.90 | 21.30 | 20.00 | 21.90 | 21.61 |
| Sm | 5.64 | 5.61 | 4.86 | 5.08 | 5.53 | 4.09 | 3.96 | 4.29 | 4.11 |
| Eu | 0.69 | 0.72 | 0.66 | 0.57 | 0.68 | 1.07 | 1.03 | 1.07 | 1.06 |
| Dy | 4.02 | 3.49 | 3.89 | 3.91 | 4.22 | 2.77 | 2.67 | 2.92 | 2.79 |
| Y | 19.40 | 17.40 | 20.70 | 19.20 | 21.90 | 15.10 | 14.60 | 15.90 | 15.20 |
| Ho | 0.76 | 0.66 | 0.77 | 0.72 | 0.83 | 0.56 | 0.55 | 0.59 | 0.57 |
| Yb | 1.87 | 1.75 | 2.05 | 1.93 | 2.19 | 1.39 | 1.38 | 1.49 | 1.42 |
| Lu | 0.31 | 0.28 | 0..34 | 0.33 | 0.37 | 0.24 | 0.24 | 0.26 | 0.25 |
| Rb | 166 | 174 | 173 | 173 | 164 | 68 | 73 | 96 | 79 |
| Ba | 260 | 260 | 270 | 200 | 250 | 510 | 500 | 510 | 507 |
| Th | 18.25 | 17.85 | 14.70 | 17.75 | 17.85 | 7.50 | 6.93 | 8.75 | 7.73 |
| U | 3.30 | 3.90 | 2.10 | 3.80 | 3.80 | 2.40 | 2.80 | 2.60 | 2.6 |
| Ta | 1.24 | 1.28 | 1.15 | 1.60 | 1.14 | 0.52 | 0.51 | 0.57 | 0.53 |
| Nb | 8.80 | 7.80 | 7.70 | 9.20 | 8.20 | 5.30 | 5.30 | 5.50 | 5.37 |
| Pb | 16.00 | 18.80 | 17.40 | 19.00 | 18.20 | 6.60 | 6.00 | 7.70 | 6.8 |
| Sr | 155 | 150 | 146 | 127 | 150 | 638 | 629 | 548 | 605 |
| P | 460 | 390 | 420 | 350 | 410 | 940 | 950 | 820 | 903 |
| Zr | 91.8 | 90.3 | 75.7 | 86.6 | 84.4 | 134.0 | 135.5 | 134.5 | 134.7 |
| Hf | 3.20 | 3.10 | 2.60 | 3.00 | 2.80 | 3.50 | 3.50 | 3.60 | 3.53 |
| Ti | 2060 | 1690 | 1800 | 1600 | 1830 | 3670 | 3750 | 3490 | 3637 |
| $\sum REE$ | 148.56 | 140.40 | 129.54 | 131.59 | 118.16 | 106.45 | 100.02 | 111.68 | 106.05 |
| $\delta Eu$ | 0.02 | 0.02 | 0.43 | 0.01 | 0.02 | 0.08 | 0.08 | 0.07 | 0.84 |
| $La_N/Yb_N$ | 9.58 | 10.89 | 7.48 | 7.70 | 8.45 | 9.43 | 9.06 | 9.66 | 9.39 |
| Rb/Sr | 1.074 | 1.163 | 1.189 | 1.367 | 1.100 | 0.106 | 0.116 | 0.174 | 0.130 |
| Sr/Y | 7.96 | 8.62 | 7.03 | 6.59 | 6.83 | 42.25 | 43.08 | 34.47 | 39.80 |

Note: Mg# = molar Mg/(Mg + Fe). Data from Cao et al. [28]. In summary, geochemical data suggest that the Zuluhong granite is a highly fractionated and weakly peraluminous I-type granite that was derived from the partial melting of lower crust.

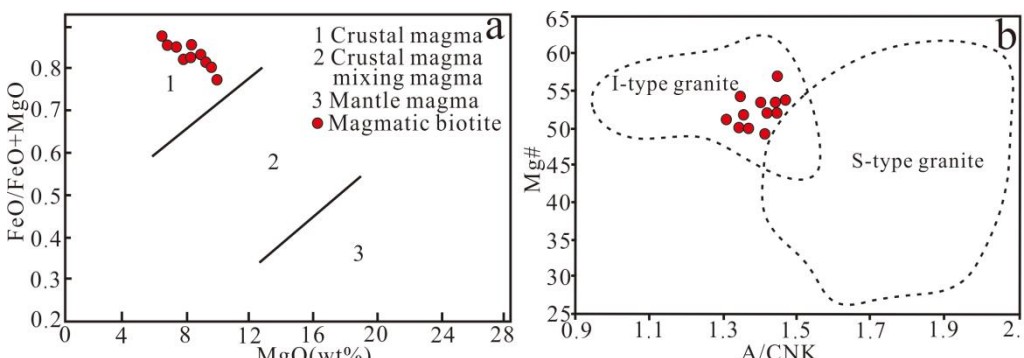

**Figure 10.** (**a**) Granite source discrimination diagram (modified after Zhou, [56]). (**b**) Biotite Mg# versus A/CNK diagram (modified after Gao, [55]).

*5.4. Implications for Tectonic Setting*

The Zuluhong monzogranite intruded the lower Carboniferous Aksha Formation and was formed during the Visean stage (346–330 Ma), according to biostratigraphic constraints. The Aksha Formation comprises a shallow-marine carbonate and clastic rock formation, typical of the sedimentary environment in a back-arc basin [57]. The zircon U–Pb age of the Zuluhong monzogranite is $320 \pm 2$ Ma [28]. The presence of the Bayingou ophiolite (325–334 Ma) [58], arc basalts (320–334 Ma) [59], and a "stitching pluton" (zircon U–Pb age of 316 Ma) in an ophiolite in northern Tianshan indicate that the western Tianshan was a complex trench–arc–basin system at ca. 320 Ma. The long period of subduction and collision in the western Tianshan region caused thickening of the lower crust, forming the Aksha Formation. Continuous stretching of a back-arc basin results in heat being transferred from the upper mantle to the crust and the development of an anomalously high geothermal gradient, which can cause melting of the lower crust.

The Paleo-Asian oceanic lithosphere was being subducted during the late Paleozoic, which may have led to crustal thickening, crustal melting, and the formation of granitic magma. The Zuluhong granite was derived from the lower crust, and its high $SiO_2$ and total alkalis are consistent with the compositions of granitic magma experimentally produced by the partial melting of basaltic rocks [60]. Hence, the Zuluhong granite may have been produced by the partial melting of basaltic rocks in the lower crust.

*5.5. Coupled W–Cu Mineralization*

There is considerable variation in the W and Cu mineralization processes, given that: (1) W is an incompatible element and enriched in continental crust, whereas Cu is a compatible element and enriched in the mantle [61]; (2) W ores are mostly quartz-vein- and skarn-type [5], whereas Cu ores are found mainly in porphyry deposits [62]; (3) W ores are generally associated with granitic magmas that are highly fractionated (high-Si) and have low oxygen fugacity [7,9,63], whereas Cu ores are typically associated with granitic magmas that have experienced little fractionation (intermediate–silicic) and have high oxygen fugacity [8,64]; and (4) W ore-forming fluids have medium–low salinity and high F content [65], whereas Cu ore-forming fluids have high salinity and high Cl (or S) content [12,13,66]. Therefore, it is unusual for both W and Cu to be present in the same ore deposit [14].

Several large W–Cu ore deposits have been recently found in China, such as at Xushan and Zhuxi in the Jiangnan orogenic belt [67,68]. The Zuluhong polymetallic deposit is also a W–Cu ore deposit and has some features that differ from those of the deposits in the Jiangnan orogenic belt. Firstly, the magma source rocks of the W–Cu deposits in the Jiangnan orogenic belt are crustal sediments and the granites are S-type granites with limited mantle input [69,70]. However, the Zuluhong monzogranite has high $SiO_2$, high alkalis, high FeO/MgO ratios (average = 4.2), high Rb/Sr ratios (average = 1.68), negative Ce anomalies, and contains accessory fluorite and tourmaline. These features demonstrate

that the Zuluhong monzogranite is a highly fractionated, I-type granite. Secondly, the ore-forming elements (i.e, W and Cu) in the deposits of the Jiangnan orogenic belt are partly magma derived [71,72], whereas the Zuluhong monzogranite has relatively low contents of the ore-forming elements [27]. Thirdly, there are several types of mineralization in the Jiangnan orogenic belt, such as altered-granite-, quartz-vein-, and veinlet–disseminated-types, and the mineralized veins are found mainly in the granite pluton [73]. However, the main type of mineralization in the Zuluhong deposit is the quartz-vein-type, and the quartz veins are found mainly in the contact metamorphic zone. As such, the Zuluhong W–Cu polymetallic deposit represents an interesting new type of W–Cu mineralization.

During alteration, both the temperature and oxygen fugacity decreased. Firstly, the temperature dropped to 209–248 °C (average = 237 °C) (the formation temperature of chlorite), and the average oxygen fugacity [log($fO_2$)] dropped from −24.3 to −41.3. The temperature and oxygen fugacity of wolframite and sulfide (chalcopyrite and pyrite) formation are different. The temperature of wolframite formation related to quartz veins is 330–380 °C (average = 350 °C) [62,74]. This temperature is slightly higher than that of the crystallization temperature of chlorite. Thus, there is only a short time interval between the crystallization of chlorite and that of wolframite. The temperature of sulfide formation ranges from 100 to 700 °C, but when sulfide crystallizes during late hydrothermal stages, the temperature is ~228 °C [75]. This temperature is close to that of chlorite formation. In magmas, W exists in the form of $W^{6+}$. A change in oxygen fugacity cannot change the valence of W in magmas or the solubility of $W^{6+}$ in the melt [76,77]. However, at high oxygen fugacities, the partition coefficient for $W^{6+}$ between melt and ore minerals increases and $W^{6+}$ tends to enter metallic minerals and form W ores [78]. Sulfur exists in the form of $S^{2-}$ in sulfides and is readily oxidized under high oxygen fugacities. As such, the formation of sulfides requires a low oxygen fugacity.

In summary, the Zuluhong ore deposit formed over at least two stages. During the first stage, granitic magma differentiated and evolved to form the early ore-forming fluids with high Si content and oxygen fugacity. These fluids precipitated wolframite in early quartz veins. During the second stage, the fluid interacted with reducing wall rocks and the temperature and oxygen fugacity decreased as biotite was altered to chlorite. Wall rock alteration in the deposit is strong, and includes greisenization, silicification, and chloritization. In the second mineralization stage, chalcopyrite and pyrite replaced wolframite or infilled voids and cleavage planes, resulting in the formation of late quartz veins.

## 6. Conclusions

1. Magmatic biotite in the Zuluhong monzogranite has an idiomorphic or hypidiomorphic texture and contains inclusions of the accessory minerals apatite and fluorite. The biotite is Fe-rich with a high Rb/Ba ratio (0.27–9.14), which indicates that the Zuluhong monzogranite is highly differentiated. Magmatic biotite has $\sum$REE contents that are 5–10% of the whole-rock REE inventory and shows slight light REE depletion and negative Ce anomalies. Magmatic biotite is also enriched in some LILE (e.g., Rb and K) and depleted in some HFSE (e.g., Nb and Ti). Based on geological constraints and geochemical data, we suggest that the Zuluhong monzogranite was derived by lower-crustal melting due to heating from asthenospheric upwelling.
2. Around mineralized quartz veins, some biotite is partially to completely altered to chlorite. Chlorite mineral chemistry record a temperature range from 209 to 248 °C and a change in oxygen fugacity [log($fO_2$)] from −24.3 to −41.3 during alteration.
3. The formation of the Zuluhong ore deposit can be divided into at least two stages: (a) differentiation and evolution of granite magma to produce the early ore-forming fluids with high Si contents and oxygen fugacity, which precipitated quartz-vein-type wolframite; and (b) fluid interaction with reducing wall rocks, biotite alteration to chlorite, and decreasing temperature and oxygen fugacity, which led to the formation of sulfide minerals such as chalcopyrite and pyrite after wolframite.

**Author Contributions:** Investigation, R.C., B.C. and S.-C.Y.; methodology and writing-original draft preparation, R.C., D.-F.C. and H.-D.G. All authors have read and agreed to the published version of the manuscript.

**Funding:** This work was supported by the Second Tibetan Plateau Scientific Expedition and Research (STEP) Programme (2019QZKK0804), the National Natural Science Foundation of China (U21A2015), and the key research and development program in Tibet (XZ202101ZY0014G).

**Data Availability Statement:** The data presented in this work are available on request from the corresponding author.

**Acknowledgments:** The authors thank Fangyue Wang for constructive comments that helped to improve the experiment. We also thank Xinghua Ma for help with thoughtful discussions and informal reviews.

**Conflicts of Interest:** The authors declare no conflict of interest.

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
