# Peer review of "Mineralogical Characteristics of Biotite and Chlorite in Zuluhong Polymetallic Deposit: Implications for Petrogenesis and Paragenesis Mechanisms of the Tungsten and Copper"

_minerals, doi:10.3390/min12101280_

Round 1

Reviewer 1 Report (Previous Reviewer 4)

During my first review, I issued a notice of "major corrections" and pointed out important shortcomings in the approaches used. The second version of the manuscript has only been modified by less than 10%, and the answers bypass the gaps. Clearly, this does not meet my expectations.

There are still no EDX maps made with EMP: we always have a figure with 5 analysis points, which is not rigorous. The Fe3+ is not measured, the d00l is not measured, everything is estimated with old methods without there being the slightest justification for not using the recent methods apart from a vague "it works". The empirical equations are used in cascade, thus propagating any errors. As proof, only one recent thermometer is cited - without being used - that of Inoue et al 2018, derived from Inoue et al 2009. When I use the chlorite data provided in the tables (also using the value of Fe3+ which is given although not measured) and apply the Inoue’s thermometer, I obtain on average 100°C more than the values ​​presented in the paper! Still, I don't think Inoue's recent semi-thermodynamic thermometer is any less reliable than the empirical equations of the 80s-90s. But there is no explanation, no comparison of methods.

I have to ask again for major corrections to be made to this paper, which as it stands does not have the scientific rigor necessary for publication in an impact factor ~3 journal.

Author Response

Thanks for your good suggestions and new suitable method.

Under the microscopic observation, we found that biotite had different optical characteristics, which maybe caused by the differences in composition of a zoned biotite. Thus, the Major element compositions analysis of different analysis points in the biotite was carried out by the electron microprobe analyzer (EMPA). Just as we expected, from the data related to electron microprobe analysis as the Figure 9 show, the MgO content increases gradually from point 1 to point 5 (Figure 9A), whereas the FeOtot content decreases (Figure 9B). The coupled decreases in MgO and TiO2 during magma evolution (Figure 9C) show that the temperature decreased during biotite crystallization and that the changes in biotite and magmatic Fe and Mg contents reflect the fractional crystallization of biotite (see lines 319-327). According to reviewer’s good suggestions, we added the reason for five analyses were conducted from the point 1 to the point 5 of a single biotite grain. (see lines 315-319) 

Major element compositions of minerals were analyzed by a JEOL JXA-8230 electron microprobe analyzer (EMPA) operated at a voltage of 15 kV, beam current of 20 nA, spot size of 1μm, and maximum counting time of 20s. However, the EMPA were unable to distinguish Fe2+ from Fe3+. Thus, we could only use the results of previous studies for extrapolation. According to reviewer’s good suggestions, we have added the comparison of methods (see lines 249-265) . Cathelineau & Nieva first reported a positive correlation between the AlVI content of chlorite and formation temperature, and proposed a solid solution thermometer for chlorite. Nieto reported a modified formula for the crystalline interplanar spacing (d001) of chlorite: d001 = 14.339 – 0.1155AlIV – 0.02011 × Fe2+. Furthermore, a mathematical relationship between d001 and the formation temperature of chlorite was proposed by Battaglia [40]: T = (14.379 – d001)/0.001. However, the reliability of this method is not ideal because the influence of whole rock composition is not considered. Moreover, the chlorite formation temperature can also be calculated using this equations proposed by Nishimoto (2010) and Maydagán (2016). Combined with the semi-empirical thermometer of Inoue and the method proposed by Walshe, Inoue has selected four chlorite components to establish a new thermodynamic model. As reviewer suggested, the method proposed by Inoue (2018) was used to calculate formation temperature of chlorite, and the data for chlorite were converted to a structural formula on the basis of 14 oxygens(see Table 3 and Line 261-265). As the Table 3 and results show, The conclusions obtained by the new method proposed by Inoue are consistent with the relevant judgment of the paper.

References:

Inoue A.; Meunier A.; Patrier-Mas P.; Rigault C.; Vieillard P. Application of chemical geothermometry to low-temperature trioctahedral chlorites. Clays and Clay Minerals. 2009, 57, 371-382.

Walshe, J.L. A six-component chlorite solid solution model and the conditions of chlorite formation in hydrothermal and geothermal systems. Economic Geology. 1986, 81, 681-703.

Inoue, A.; Inoue, S.; Utada, M. Application of chlorite thermometry to estimation of formation temperature and redox conditions. Clay Minerals. 2018, 53, 2, 143-158.

Fuat Y.; Mustafa K.; Necati K.; Muazzez Ç. K.; Demet K. Y. A Windows program for chlorite calculation and classification. Computers & Geosciences. 2015, 81, 101-113.

Reviewer 2 Report (Previous Reviewer 3)

I have reviewed this article before (No:1846876), and the authors revised it according to the last opinion. But there is still one problem in this manuscript that needs to be revised. The whole rock major and trace elements of this paper have been published in the author's another article on Lithos. The authors should clarify this issue in the main text. Data cannot be republished again.

 Cao, R.; Bagas, L.; Chen, B.; Wang, Z.Q.; Gao, Y.B. Geochronology and petrogenesis of the composite Zuluhong Granite, North Xinjiang Province of China: Implications for the crust-mantle interaction and continental crustal growth in Western Tianshan Orogen. Lithos. 2021, 380–381, 1-16.

Author Response

Thanks, we have added the corresponding references. (see Line 372)

Reviewer 3 Report (Previous Reviewer 1)

Dear authors,

Comparing this version with previous versions, I believe that the manuscript has been improved and that the manuscript warrants publication in minerals. However, I have detected some typographical errors, which are listed below. I have pointed them out throughout the PDF and in a document, I have attached. I also suggest replacing one sentence so that no doubts arise in the reader. The article can be accepted for publication after some revisions.

General comments:

1.      References: please see “MDPI Reference List and Citations Style Guide” in  https://www.mdpi.com/authors/references. See the document: “Download the full MDPI Reference List and Citations Style Guide (PDF, 520KB)”. References to papers in the text should omit the date (example: Line 58 – write “Yuguchi et al. [25]” in place of  “Yuguchi et al. (2015) [25]”).  

2.      Some typo/ formatting errors were identified throughout the manuscript. Please see with careful attention everything that is underlined in yellow in the manuscript PDF.

Best regards 

Author Response

  1. References: please see “MDPI Reference List and Citations Style Guide” in  https://www.mdpi.com/authors/references. See the document: “Download the full MDPI Reference List and Citations Style Guide (PDF, 520KB)”. References to papers in the text should omit the date (example: Line 58 – write “Yuguchi et al. [25]” in place of  “Yuguchi et al. (2015) [25]”).  

Our Reply: Thanks, we have revised it as reviewer suggested.

  1. Some typo/ formatting errors were identified throughout the manuscript. Please see with careful attention everything that is underlined in yellow in the manuscript PDF.

Our Reply: Thanks, we have revised it as reviewer suggested.

Round 2

Reviewer 1 Report (Previous Reviewer 4)

This third version of the paper entitled "Mineralogical characteristics of biotite and chlorite in Zuluhong polymetallic deposit: Implications for petrogenesis and paragenesis mechanisms of the tungsten and copper" is a good new version.

I appreciate that Authors used a recent chlorite thermometric tool, taking into account the bulk rock chemistry, and giving a semi-thermodynamic base to the T estimates. I also appreciate that authors removed the Fe3+ content from chlorite, which was initially deduced by stoichiometry, with all the problems that this raises (especially for thermometric purposes).

However, I stand by my position regarding the Figure 8. This one does not do justice to the article. The paper will be greatly improved if authors agree to insert a real EMP map, instead of fig.8/fig.9.

Author Response

Thanks. According to reviewer’s suggestions, we have added the EDX maps of biotite (see Figure 8b). The EDX map of the biotite was collected at the School of Earth Sciences, Chengdu University of Technology, Chengdu, China, using a Nova 450 field-emission scanning electron microscope at a voltage of 30 kV (see line 125-127).

This manuscript is a resubmission of an earlier submission. The following is a list of the peer review reports and author responses from that submission.

Round 1

Reviewer 1 Report

Dear authors,

The manuscript has been improved and I believe the manuscript warrants publication in minerals. However, I have detected some typographical errors, which are listed below. The article can be accepted for publication after some revisions.

- Line 72 and others – The " dot" in “Figure .1A” (and others)  is not necessary. You can just put it (Figure 1A). Valid for all references to figures throughout the text.

- Line 121 – Please use "capital letter” at the beginning of the sentence: "3. Analytical methods".

- Lines 121 to 134 - I think that paragraph between lines 121 and 134 is misplaced. That paragraph is either still about section "2. Regional geological background and mineralization" or it is already “4. Results”. Maybe in another subtopic about “4.1. Sample description” or “ 4.1. Petrographic description”. In my opinion it fits better under Results.

- Line 140 - The section "3. Analytical methods" should be started before line 140, before "Ten thin sections of the Zuluhong ..."

- Line 161 to 168 + 206 to 208 + 355 to 356 – the tables are unformatted. Please confirm.

- Lines 353 and 354 - AlVI (the “VI” should be in superscript).

Thank you,

Best regards.

Author Response

Comments from Reviewer #1:

The manuscript has been improved and I believe the manuscript warrants publication in minerals. However, I have detected some typographical errors, which are listed below. The article can be accepted for publication after some revisions.

Comments 1: - Line 72 and others – The " dot" in “Figure .1A” (and others) is not necessary. You can just put it (Figure 1A). Valid for all references to figures throughout the text.

Our Reply: Thanks, We have deleted the " dot" in Figure 1 and other Figures.

Comments 2: - Line 121 – Please use "capital letter” at the beginning of the sentence: "3. Analytical methods".

Our Reply: Yes, we have revised "3. analytical methods" as "capital letter". (see line 125, p. 4)

Comments 3: - Lines 121 to 134 - I think that paragraph between lines 121 and 134 is misplaced. That paragraph is either still about section "2. Regional geological background and mineralization" or it is already “4. Results”. Maybe in another subtopic about “4.1. Sample description” or “ 4.1. Petrographic description”. In my opinion it fits better under Results.

Our Reply: Yes, we have revised it as suggested. We moved “petrographic description” to the chapter 4 as a specific paragraph.(see line 105-127, p. 5)

Comments 4: - Line 140 - The section "3. Analytical methods" should be started before line 140, before "Ten thin sections of the Zuluhong ..."

Our Reply: Yes, we have moved the "3. Analytical methods" in front of "Ten thin sections of the Zuluhong ...". (see line 125-142, p. 4)

Comments 5: - Line 161 to 168 + 206 to 208 + 355 to 356 – the tables are unformatted. Please confirm.

Our Reply: Thanks, We have revised the format of the table.(see table 1-4)

Comments 6: - Lines 353 and 354 - AlVI (the “VI” should be in superscript).

Our Reply: Thanks, we have revised the “VI” as superscript.(see line 373-374)

Reviewer 2 Report

I have checked the revision notes of MS#1846876, which is quite similar to that of MS#1775605, with no significant improvement. Particularly, in my previous review, I suggest the authors use a new method (Li et al., Lithos 2020), to calculate formula, which is important for their discussion using Fe2+/Fe3+ of biotite; however, the authors insisted on using an old method that was published in a Chinese local journal without an intentional review. It is strange that the authors claim in their reply that these two methods give similar biotite formula, but they did not provide results from both methods to prove that. I believe that results would be quite different. In addition, the authors claim the the Zuluhong monzogranite is highly fractionated and F-rich, but they also claim that F in the biotites are too low to be measured; it cannot be both true. Conclusively, I believe that authors did not try their best to revise this manuscript, and I recommend rejection again. 

Author Response

Comments from Reviewer #2:

Comments 1: I have checked the revision notes of MS#1846876, which is quite similar to that of MS#1775605, with no significant improvement. Particularly, in my previous review, I suggest the authors use a new method (Li et al., Lithos 2020), to calculate formula, which is important for their discussion using Fe2+/Fe3+ of biotite; however, the authors insisted on using an old method that was published in a Chinese local journal without an intentional review. It is strange that the authors claim in their reply that these two methods give similar biotite formula, but they did not provide results from both methods to prove that. I believe that results would be quite different. In addition, the authors claim the the Zuluhong monzogranite is highly fractionated and F-rich, but they also claim that F in the biotites are too low to be measured; it cannot be both true. Conclusively, I believe that authors did not try their best to revise this manuscript, and I recommend rejection again.

Our Reply: Lin et al(1994) has reviewed the literature and related summary on the composition analysis and calculation of biotite; many scholars have adopted the calculation method proposed by Lin et al(1994) and published in relevant famous SCI journals, such as Zheng et al(Acta petrologica sinica, 2022), Zhang et al(Science China-earth sciences, 2013); therefore, the calculation method by Lin et al(1994) is feasible and typical method. Furthermore, our biotite data meets the applicable conditions of the calculation method proposed by Lin, therefore, we chose the calculation method of Lin. Moreover, thanks to the reviewer for providing other calculation methods, we have carried out the calculation of biotite composition and found that the calculation results are similar to those of method proposed by Lin et al(1994). The method proposed by Li has no effect on the relevant conclusions of the paper. Therefore, we insist on using the method proposed by Lin.

According to our study, the Zuluhong Granite are characterised by its high SiO2 content (72.7–74.4 wt%), moderate 10000×Ga/Al values(2.59–2.81), strongly negative Eu anomaly(0.39–0.43), and relatively low zircon saturation temperatures(Tzr) of 726–807℃, which are indicative of a highly fractionated granite. We judged initially the probable reason for the highly fractionated granite is the F-rich magma. Unfortunately, we found that it is not rich in fluorine in the whole rock analysis and that fluorine is not detected in biotite. Therefore, it could be other reasons for the highly fractionated Zuluhong granite other than F-rich magma. We accept the suggestions of the reviewer and delete the relevant expression of F-rich granite in the article.

References:

Cao, R.; Bagas, L.; Chen, B.; Wang, Z.Q.; Gao, Y.B. Geochronology and petrogenesis of the composite Zuluhong Granite, North Xinjiang Province of China: Implications for the crust-mantle interaction and continental crustal growth in Western Tianshan Orogen. Lithos. 2021, 380–381, 1-16.

Allen, M., Windley, B., Zhang, C., 1992. Palaeozoic collisional tectonics and magmatism of the Chinese Tien Shan, Central Asia. Tectonophysics 220 (1), 89–115.

Chen, B., Jahn, B.M., 2004. Genesis of post-collisional granitoids and basement nature of the Junggar Terrane, NW China: Nd-Sr isotope and trace element evidence. J. Asian Earth Sci. 23 (5), 691–703.

Du, L., Long, X.P., Yuan, C., Zhang, Y.Y., Huang, Z.Y., Wang, X.Y., Yang, Y.H., 2018. Mantle contribution and tectonic transition in the Aqishan-Yamansu Belt, Eastern Tianshan, NW China: Insights from geochronology and geochemistry of early Carboniferous to early Permian felsic intrusions. Lithos 304–307, 230–244.

Zhang Xiao-Li, Hu Ling, Ji Mo, Liu Jun-Lai & Song Hong-Lin. Microstructures and deformation mechanisms of hornblende in Guandi complex, the Western Hills, Beijing. Science china-earth sciences. 2013, 56(9), 1510-1518.

Zheng Y L, Zhao Z, Zhang C Q, Li H W, Li B. Genetic relationship between the two-period magmatism and tungsten mineralization in the Yangchuling deposit, Jiangxi Province: Evidence from biotite geochemistry. Acta petrologica sinica. 2022, 38(2), 495-512.

Lin, W W; Peng, L J. The Estimation of Fe3+ and Fe2+ contents in amphibole and biotite from EMPA data. Journal of Jilin University(Earth Science Edition). 1994, 24, 155-162.

Reviewer 3 Report

In this paper, the diagenetic process of magmatic rock and the metallogenic process of tungsten-copper deposit are studied by using biotite and chlorite.  The text is smooth, the pictures are beautiful, the data is sufficient and the discussion is in-depth.  I suggest a minor revision.  However, the following minor issues need to be corrected in this article. 

(1) The data of major, trace and rare earth elements of Zhuluhong granite in Fig. 5 and Table 4 need to be written clearly.  Was it tested in this study?  Test methods need to be clearly written in the text. 

(2) In this study, the trace and rare earth elements of altered biotite and chlorite were not tested in situ.  Why is that? 

(3) Lines 353-354. VI needs superscript. 

(4) Line 4 Copper 

(5) The content expressed in Figure 1A is unclear.

Author Response

Comments from Reviewer #3:

In this paper, the diagenetic process of magmatic rock and the metallogenic process of tungsten-copper deposit are studied by using biotite and chlorite.  The text is smooth, the pictures are beautiful, the data is sufficient and the discussion is in-depth.  I suggest a minor revision.  However, the following minor issues need to be corrected in this article.

Comments 1: The data of major, trace and rare earth elements of Zhuluhong granite in Fig. 5 and Table 4 need to be written clearly.  Was it tested in this study?  Test methods need to be clearly written in the text.

Our Reply: We have carried out the experiment and analysis of major and trace elements of the whole-rock of Zuluhong granite, and the relevant results have been published in Lithos. This study focuses on the micro analysis of single minerals such as biotite and chlorite of Zuluhong granite, and compares with the major and trace elements of the whole-rock of Zuluhong granite. This study investigates the magma source and evolution, fluid properties, and oxygen fugacity changes during alteration, through major and trace element analyses of magmatic and altered biotite in the Zuluhong deposit. As the reviewer suggested, in Figure 5, we supplemented the data sources of major and trace elements related to the whole rock of Zuluhong granite.

Comments 2: In this study, the trace and rare earth elements of altered biotite and chlorite were not tested in situ. Why is that?

Our Reply: Yes, the trace and rare-earth elements of magmatic biotite are compared with those of the whole-rock to discuss the genesis of Zuluhong granite. Furthermore, the trace elements and rare-earth elements of partially altered biotite and chlorite lack consistency, which is not a good indicator of the genesis of Zuluhong granite. Therefore, in this study, we mainly used the major elements of chlorite and biotite to analyze the oxygen fugacity of their formation and then characterize the mineralization physicalchemical condition.

Comments 3: Lines 353-354. VI needs superscript. 

Our Reply: Thanks, we have revised the “VI” as superscript.(see line 373-374)

Comments 4: Line 4 Copper 

Our Reply: Yes, we have revised it as suggested.(see line 4)

Comments 5: The content expressed in Figure 1A is unclear.

Our Reply: Thanks, We already have redrawn Figure 1 to ensure it is clear.(see Figure 1)

Reviewer 4 Report

The paper untitled « Mineralogical characteristics of biotite and chlorite in Zuluhong polymetallic deposit: implications for petrogenesis and paragenesis mechanisms of the tungsten and copper” focuses on the composition and evolution of biotite (chloritization) in a granite from China which has the particularity of presenting a tungsten and copper coexistence.

This paper would be an interesting paper if there weren't so many big gaps in the methods. Many results are presented without giving any indication of how they were obtained. Other methods are used, but not in a relevant way. Finally, this paper suffers from numerous repetitions or evidence, as well as approximations.

The authors do not specify how many analyzes were made, nor how they were done (in tables, all crystal core analyses?), nor how they were selected (contaminated analyses?).

Authors present data of compositions based on a Fe2+/Fe3+ ratio. Where do these results come from? How were they obtained? Was measured? We hope so! But how? Was estimated? Unfortunately, most likely, and one wonders how, especially for chlorite, since even without Fe3+ the structural formula is consistent.

In figure 8 – which has no scale - it is not possible to see the whole range of biotite to really see the location of the analyses. Is the core analysis right in the middle? Above all - the most important point - when we use an EMP and we want to highlight an intracrystalline chemical zonation, we do an EDX map. Microprobes make maps. Not just 5 points and an approximate dotted line which do not allow to apprehend the evolution of the zonation. And all the more so since the range of biotite which is shown is itself made up of crystallites of different orientation.

The chlorite thermometry is another example of inappropriate use of methods. Or inappropriate methods. Rather than using the chemical composition of chlorites to estimate the T°C, or doing XRD analyzes to measure the d001 interplanar spacing, the authors use chlorite composition to calculate a d001 spacing, to then use the latter to calculate a temperature... risking accumulating approximations, especially considering that the equation reported line 246 as the Nieto’s equation is not well written (AlIV and not AlVI, 0.1155 and not 0.115, …). Moreover, it does not seem that the calculated d001 presented in table 3 are in agreement with this equation. This makes no sense as many direct methods are available in the literature.

Moreover, the empirical equations used are from the 80s and 90s, without ever saying why this one rather than another (there are more than 10 chlorite thermometric equations in the literature to do this). Above all, the authors put aside all the recent work on chlorite thermometry, which is nevertheless much more reliable than the succession of empirical equations (the result of which depends on the geological field of application and the chosen equation): Vidal et al., 2002, 2005, 2006, 2016; Inoue et al., 2009, 2010, 2018 (this later allowing the oxygen fugacity calculation); Bourdelle et al., 2013, Bourdelle and Cathelineau 2015, Bourdelle 2021….

By the way, generally the literature used by the authors is aging. Average publication date of citations: 2001.1 (more than 20 years). 75% are more than 10 years old.

Approximations

Line 100-101 “The Zuluhong Granite can be divided into a monzogranite and porphyritic granodiorite from its core to rim.” Why this division does not appear on the map (Fig. 1)? Unless the monzogranite is the granite of the map, and the granodiorite the diorite in orange? Not clear at all.

Above all, line 141, “Monzogranite […] The main rock-forming minerals are quartz (30%–35%), alkali feldspar (15%–20%), and biotite (~8%). No hornblende or muscovite are present. Accessory minerals (~2%)…”. 30+15+8+2 = 55%. And the rest? 45% of plagioclase? But in this way, it is a granodiorite, not a monzogranite.

Lines 149-151. If biotites are completely replaced by chlorite, then they are no longer biotites. It is the chlorites that are green.

Lines 257-269. The authors speak of "replacement" of one layer by another, say that the K content decreases with biotite alteration - which does nothing, chloritization leads to a phase that does not contain any - etc. This is all unclear, suggesting a SST transformation while it is a hydrothermal alteration more favorable to DC (see the work of Putnis). In fact the mechanisms aren't really described, and there are no references.

Lines 270-280. Must be rewritten because this paragraph brings nothing.

Minor modifications

Line 160. Fig. 1, 2 and 3.

Line 126. EMP spot size of 1mm? Rather 1 µm.

Fig. 1: unclear maps. (b) too small. The names of the samples are missing.

Fig. 2 (a) and (c): we see nothing.

Line 236. Reference [33], not [34]

Round 2

Reviewer 2 Report

First, the authors insisted on using the old-fashioned method from the chinese journal, although I have recommened them to use an updated one recently published on Lithos; I do not know why, provided that we take Minerals as an international journal. Instead, the authors claim that some recently published works still use the method from that old reference, but I do not see the logic here (why do you follow bad examples?). Second, many data of so-called partially altered biotites (labled as P**) in Table 1 have very low SiO2 and extremely low K2O, these compositions cannot be biotite but most likely chlorite, and therefore the “partially altered biotites” cannot be plotted in diagrams designed for biotite (e.g., Fig. 4a, Fig. 6).